behaviour, health and disease and epidemiology, evolution

social immunity, clonal raider ant, automated tracking, interaction network, social behaviour

**Author for correspondence:**
Yuko Ulrich
e-mail: yulrich@ice.mpg.de

# Immune challenges increase network centrality in a queenless ant

Giacomo Alciatore[1,2], Line V. Ugelvig[3,4], Erik Frank[2,5], Jérémie Bidaux[2], Asaf Gal[3], Thomas Schmitt[5], Daniel J. C. Kronauer[3] and Yuko Ulrich[2,6]

[1]Institute of Integrative Biology, ETHZ Zürich, 8092 Zürich, Switzerland
[2]Department of Ecology and Evolution, University of Lausanne, 1015 Lausanne, Switzerland
[3]Laboratory of Social Evolution and Behavior, The Rockefeller University, New York, NY 10065, USA
[4]Centre for Social Evolution, University of Copenhagen, 2100 Copenhagen N, Denmark
[5]Department of Animal Ecology and Tropical Biology, University of Würzburg, Biocentre, 97074 Würzburg, Germany
[6]Max Planck Institute for Chemical Ecology, Hans-Knöll-Straße 8, 07745 Jena, Germany

GA, 0000-0001-5626-728X; LVU, 0000-0003-1832-8883; EF, 0000-0002-2066-3202;
AG, 0000-0003-0834-2649; TS, 0000-0002-6719-8635; DJCK, 0000-0002-4103-7729;
YU, 0000-0001-9380-7863

Social animals display a wide range of behavioural defences against infectious diseases, some of which increase social contacts with infectious individuals (e.g. mutual grooming), while others decrease them (e.g. social exclusion). These defences often rely on the detection of infectious individuals, but this can be achieved in several ways that are difficult to differentiate. Here, we combine non-pathogenic immune challenges with automated tracking in colonies of the clonal raider ant to ask whether ants can detect the immune status of their social partners and to quantify their behavioural responses to this perceived infection risk. We first show that a key behavioural response elicited by live pathogens (allogrooming) can be qualitatively recapitulated by immune challenges alone. Automated scoring of interactions between all colony members reveals that this behavioural response increases the network centrality of immune-challenged individuals through a general increase in physical contacts. These results show that ants can detect the immune status of their nest-mates and respond with a general 'caring' strategy, rather than avoidance, towards social partners that are perceived to be infectious. Finally, we find no evidence that changes in cuticular hydrocarbon profiles drive these behavioural effects.

## 1. Introduction

Social animals are particularly vulnerable to infectious diseases because spatial proximity, frequent social interactions and shared resources can facilitate the spread of pathogens [1–3]. Many social species have evolved physiological, behavioural and organizational adaptations to counteract this risk [2,4,5]. Among social species, social insects display a particularly rich repertoire of group-level anti-pathogenic defences, collectively termed 'social immunity' [6,7], including mutual grooming (allogrooming) [8,9], waste management [10], disinfection of the nest with antimicrobial substances [11,12] and global patterns of social interactions that reduce disease transmission (organizational immunity) [13–15]. Allogrooming is a widespread behaviour in social insects and plays a key role in pathogen defence: by effectively removing infectious particles from, or spreading antiseptic substances onto the body surface of nest-mates [8], this behaviour can increase the survival of pathogen-exposed individuals [16–18]. However, because it involves physical contact, it can also lead to pathogen transmission [17,19]. An alternative or additional way to contain disease spread is to limit opportunities for disease transmission by reducing social interactions between infectious individuals and their nest-mates [13,20,21]. In particular, theoretical

epidemiology predicts that various properties of interaction networks (i.e. group-level) and of the nodes within these networks (i.e. individual-level) can reduce global disease spread and individual infection risk [22,23]. Some of these predictions are supported by empirical studies showing, for example, that lower network density [24] and low node strength and centrality [25] decrease transmission, or that pathogen exposure leads to changes in network properties that are predicted to slow disease transmission [14]. While the efficacy of both allogrooming and changes in global interaction patterns as anti-pathogenic defences have been demonstrated [14,17,18], these strategies have seemingly opposite effects on the interaction frequency of infectious individuals. Intuitively, allogrooming increases social interactions, while organizational immunity generally predicts decreased social interactions [13]. Therefore, the decision to either care for infectious nest-mates or avoid them is often thought to be context-dependent and adjusted to risk [26,27]. Alternatively, a colony could in principle gain the benefits of allogrooming while limiting the colony-wide transmission risk associated with increased social contacts by skewing the social interactions of infectious colony members towards a reduced set of social partners (e.g. specialized carers). However, investigating this question requires the ability to record interactions occurring between all colony members, which is still practically challenging in most systems.

Irrespective of the behavioural response elicited by the presence of infectious nest-mates, a response first requires the ability to detect an infection risk. This can be achieved, for example, through the detection of the infectious agents themselves [9,28,29]. This requires the ability to detect different pathogens and to reliably distinguish them from non-pathogenic or beneficial organisms. Alternatively, infection risk can be detected through the phenotype of social partners, for example, through cues tied to those partners' immune responses to infection. For instance, experimental increases in immune activity (in the absence of pathogens) can be detected through visual [30] or chemical cues [12,15,31–35]. In principle, detecting the immune status of social partners may constitute a more robust and general detection mechanism (e.g. allowing responses to pathogens encountered for the first time, or to internal pathogens that cannot be detected directly), although it may be slower because of the time necessary to mount an immune response.

Here, we investigate the behavioural responses of ant colonies to experimental changes in the immune status of individual colony members. To manipulate the immune status of individual ants, we use non-pathogenic immune challenges (injections with immune elicitors). By simulating an infection, immune challenges allow us to study the effects of immune activation while ruling out the effects of a pathogen's virulence, manipulation of host behaviour and transmission to other colony members [3,36]. While the effects of immune challenges with various immune elicitors on the insect immune system are well characterized [37–42], few studies have quantified their effects on behaviour in social insects [15,31,33,35,43,44], and the reported behavioural effects are based on focal-individual approaches that provide limited information on colony-level responses. Here, we use automated tracking to measure both individual behaviour and patterns of interactions between all colony members. This allows us to evaluate how immune challenges affect the network position of challenged individuals and the colony's network structure over time.

To quantify the effects of immune challenges on behaviour while controlling other sources of behavioural variation, we use a social insect affording a high degree of experimental control, the clonal raider ant *Ooceraea biroi*. Colonies of this species are naturally queenless and consist exclusively of workers that all reproduce asexually (by thelytokous parthenogenesis) and synchronously, thereby producing discrete cohorts of genetically and morphologically nearly identical workers. This allows us to minimize variation in several factors known to affect behaviour in social insects (e.g. age, genotype, morphology) both within and between experimental colonies [45].

Social immunity is thought to be promoted by two characteristics of most social insects: (i) reproductive division of labour between reproductive queens and non-reproductive workers, which creates large asymmetries in value across colony members, so that queens are shielded from infectious disease by their more dispensable workers [7,27]; and (ii) high relatedness among colony members, which promotes such altruistic behaviour [6]. In the clonal raider ant, however, relatedness is maximal, but there is no strict reproductive division of labour [46]. Because social immunity has not yet been reported in ants lacking reproductive division of labour, we first use pathogenic infections to demonstrate that *O. biroi* displays a form of social immunity that is qualitatively similar to that of other social insects. We then combine manual and automated behavioural analyses to assess whether immune challenges alone can induce one important component of social immunity, allogrooming and how that, in turn, affects the network position of challenged individuals.

Finally, we explore putative cues driving behavioural responses to immune challenges. We focus on cuticular hydrocarbons (CHCs), non-volatile compounds carried on the body surface that plays a key role in social insect communication [47]. Immune challenges can alter the CHC profiles of honeybees [31–33] and ant brood [12], suggesting a role for CHCs in signalling immune status. Because social interactions *per se*—including allogrooming [48,49]—can alter CHC profiles, changes in CHCs observed in experimental social groups could result from a direct effect of the experimental treatment on CHCs, or a change in social interactions induced by the treatment. To investigate the effects of immune challenges on CHCs while controlling for any effect of social interactions, we compare CHC profiles of individuals kept in groups or alone, i.e. with or without social interactions.

## 2. Material and methods

### (a) Pathogen exposure experiments

#### (i) Pathogen exposure

We exposed ants to the generalist entomopathogenic fungus *Metarhizium robertsii* strain ARSEF 2575. Contact with the host cuticle induces *M. robertsii* conidiospores to germinate and penetrate the cuticle, typically within 24–48 h [50,51]. Thereafter, the fungus multiplies and spreads within the haemocoel, which at high conidiospore exposure doses results in host death within 3–7 days. The fungus then breaches the cuticle to form conidiospores on the cadaver's surface, which can infect new hosts. For pathogen exposure, we used a suspension of $9 \times 10^7$ conidiospores ml$^{-1}$ in sterile 0.05% Triton X-100. Prior to exposure, the conidiospore germination rate was assessed to be greater than 95% by inoculating a standard Sabouraud dextrose agar plate and incubating at 23°C for 20 h. Ants were immersed in 100 µl conidiospore suspension for 5 s and left on a filter paper for 10 min before being transferred to experimental colonies to assess survival

(see 'Survival assay') or behaviour (see 'Behavioural assay'). The same procedure was used to sham-expose ants, using sterile 0.05% Triton X-100 without conidiospores. Ants were treated in batches of eight individuals and batch was subsequently included in statistical analyses (see 'Statistical analyses').

### (ii) Survival assay

To assess the effect of pathogen exposure on mortality, the survival of pathogen-exposed and sham-exposed ants was monitored daily for 13 days post-exposure. Ants were kept in Petri dishes (Ø: 3 cm) either alone or in groups with four naive nest-mates, using 60 replicates for each combination of exposure and social environment. Half of the treated ants were young (30 days old) and the other half old (210 days old). On the day of death, cadavers were surface-sterilized, moved to a Petri dish with a damp filter paper, and observed for 14 days to track fungal outgrowth [52]. All ants belonged to clonal genotype A. Henceforth, 'genotype' refers to distinct mitochondrial haplotypes [53]. Genotype choice was based on the availability of newly eclosed cohorts of workers at the time each experiment was performed.

### (iii) Behavioural assay

To quantify behavioural responses to pathogen-exposed nest-mates, 12 colonies were set up in three-chamber (chamber Ø: 3 cm) acrylic nests with a damp plaster floor. Each colony contained two pathogen-exposed and two sham-exposed ants (one young and one old ant for each treatment; young and old as above), 12 naive ants (two young, two old and eight 90 days old ants; young and old as above) and eight pupae. Previous work has shown that colonies of $ca$ 10 workers have high fitness and display normal group-level behaviour in this system [54,55]. All adults and pupae belonged to genotype A [53]. Adults were individually colour-marked with paint on the thorax and gaster. Videos were recorded from all colonies and grooming received by pathogen- and sham-exposed ants was quantified manually during the first 10 min of each 6 h period in the 2 days following exposure. Individuals that could not be identified in a video were excluded from the analysis at the corresponding time point.

### (iv) Statistical analyses

All statistical analyses were carried out in R v. 3.5.2. All bootstrapped confidence intervals were generated using 1000 iterations. Survival was analysed using a Cox proportional hazard regression mixed model for censored data (function $coxme$ from package $coxme$). We evaluated the significance of fixed effects and their interaction by comparing models using log-likelihood ratio tests following the deletion of terms (starting with interactions). Age was originally included as a fixed factor in the model, but as it did not significantly affect survival ($\chi^2_1 = 0.77$, $p = 0.380$), it was removed from the model, and variation in age was not included in subsequent experiments. In the final model, we used treatment (pathogen- versus sham-exposed), social environment (alone versus grouped) and their interaction as fixed factors, and infection batch as a random factor. To compare survival between pairs of treatments, we conducted pairwise log-rank tests (function $pairwise\_survdiff$ from package $survminer$), using the Benjamini–Hochberg adjustment for multiple testing. Received grooming (in seconds) was analysed using a Tweedie generalized linear mixed model (GLMM, function $glmmTMB$ from package $glmmTMB$) with treatment (pathogen- versus sham-exposed), time post-exposure (a nine-level factor), and their interaction as fixed factors, and individual nested in colony as a random factor. Model assumptions were verified using the function $simulateResiduals$ from package $DHARMa$. Additionally, for models with a significant interaction between treatment and time, pairwise comparisons between pathogen-exposed and

sham-exposed ants at each time post-exposure were conducted (function $emmeans$ from package $emmeans$).

## (b) Immune-challenge experiments

### (i) Immune challenges

To increase immune activity in individual ants, we used β-1,3-glucans from S$accharomyces cerevisiae$ cell walls contained in Zymosan-A (Sigma-Aldrich), a known elicitor of the insect immune system [39,40]. A solution of 10 mg ml$^{-1}$ of Zymosan-A in phosphate-buffered saline (PBS) was vortexed for 1 h, centrifuged at 14 000 r.p.m. for 5 min and the supernatant was used for injections. Individual workers were injected under the largest abdominal tergite with approximately 0.1 µl of supernatant (immune-challenged) or PBS (control-injected) using a 36 gauge bevelled needle attached to a NanoFil syringe (World Precision Instruments), following [56]. Control-injected and immune-challenged ants had similar survival (20 versus 18 out of 24 ants survived 48 h post-injection; two-proportions $z$-test: $\chi^2 = 0.13$, $p = 0.722$), consistent with the expectation that immune challenges do not increase mortality.

### (ii) Behavioural assays

To assess the effects of immune challenges on individual-level and colony-level behaviour, we set up 24 colonies containing one immune-challenged ant, one control-injected ant (both 60 days old) and seven naive ants (unknown age), as well as five larvae. All adult ants and larvae belonged to genotype B [53]. Adults were individually marked as above. Experimental colonies were hosted in Petri dishes (Ø: 5 cm) with a damp plaster floor. Grooming received by immune-challenged and control-injected ants was manually scored from video recordings for 10 min every 6 h from 6 to 54 h post-injection (i.e. 90 min in total per colony) using the software BORIS 5.1.0.

Additionally, the software anTraX [57] was used to extract the trajectories of all ants continuously from 6 to 54 h post-injection. For each ant, the following behaviours were quantified in time windows of 6 h (except for the first and last time windows, which were 3 h long): (i) isolation, defined as the proportion of time in which an ant was greater than 1 mm ($ca$ half a body length) away from any other ant; (ii) activity, defined as the proportion of time in which the ant was moving at greater than 1 mm s$^{-1}$; and (iii) mean walking speed, in mm s$^{-1}$.

We also built networks of physical contacts between all colony members in each time window. Network nodes (individuals) were linked by weighted edges representing the duration of contact between each pair of ants. Two ants were considered to be in physical contact if the centroids of their segmented silhouettes were less than 1 mm away from each other. As clonal raider ants are 2–3 mm long, this cut-off ensured that segmented silhouettes were touching. However, segmented silhouettes can also contain more than two ants, in which case the physical contact between pairs of ants is not necessarily guaranteed. Thus, to validate our approach (i.e. to ensure that the 1 mm cut-off accurately reflected pairwise physical contacts), we manually scored all contacts in four 10 min videos from two colonies (at 18 and 48 h post-injection) using the software BORIS 5.1.0, and computed Pearson correlations between total contact durations between all pairs of ants obtained by automated and manual scoring. We calculated two node centrality measures, node strength and eigenvector centrality, to quantify the 'spreading influence' of immune-challenged, control-injected and naive ants within their social network. The node strength was computed as the sum of a node's weighted edges over 1 h and represents the mean hourly contact time of an individual in each time window [58]. The eigenvector centrality is a relative score taking into account both the connections to other nodes and the identity of those nodes, so that connections to high-scoring nodes contribute more to a node's

eigenvector centrality than equal connections to low-scoring nodes [59]. We also defined skewness of contacts as the weight of an individual's strongest edge divided by the summed weights of all its edges. This metric ranges between $1/n$ when the focal individual is in contact with its $n$ nest-mates equally and 1 when the focal ant is in contact exclusively with one nest-mate. For a visual representation, we calculated the mean node strength and mean contact skewness across all naive ants in each colony and selected the eigenvector centrality of a random ant in each colony (the mean of eigenvector centrality is not informative). When either the immune-challenged or the control-injected ant died in a colony, data from that colony were excluded from analyses from that time point onwards.

### (iii) Cuticular hydrocarbon profiles

To assess whether immune challenges affect CHCs, we compared the CHC profiles of 20 immune-challenged, 20 control-injected (injection procedures as above) and 20 unmanipulated naive ants. All ants belonged to genotype D [53] and were 60 days old. Ants were hosted in Petri dishes (Ø: 5 cm) with a damp plaster floor either alone (10 ants per treatment) or with five nest-mates of the same genotype (10 ants per treatment). All ants were frozen at −80°C 18 h post-injection. CHCs were extracted from individual workers by whole-body immersion in 1 ml hexane for 10 min. After adjusting extracts to a volume of *ca* 15 µl, 1 µl was analysed by gas chromatography-mass spectrometry (GC-MS) on a 6890 gas chromatography coupled to a 5975 mass selective detector (Agilent Technologies). The GC was equipped with a DB-5 capillary column (30 m × 0.25 mm internal diameter; film thickness 0.25 µm; J&W Scientific). Helium was used as a carrier gas with a constant flow of 1 ml min$^{-1}$. A temperature programme increasing by 5°C min$^{-1}$ from 60 to 300°C followed by 10 min at 300°C was used, with data collection starting 4 min post-injection. Mass spectra were acquired with an ionization voltage of 70 eV electron ionization-MS and a source temperature of 230°C. The software CHEMSTATION G1701AA v. A.03.00 (Agilent Technologies) was used for data analysis. Compounds were identified using diagnostic ions and identifications were confirmed using retention indices [60]. Samples that showed signs of contamination or low-quality extraction were discarded.

### (iv) Statistical analyses

Received grooming was analysed as above (Pathogen exposure experiments', Statistical analyses). To evaluate determinants of individual behaviour (isolation, activity, mean speed) and node parameters (eigenvector centrality, strength, skewness), linear mixed-effects models (LMM) were fitted (function *lme* from package *nlme*) with treatment (immune-challenged versus control-injected), time post-injection (9-level factor), and their interaction as fixed effects, and individual nested in colony as a random factor (to account for the non-independence of ants from the same colony [61]). Node strength was log-transformed to satisfy model assumptions. Variance structures for both treatment and time were included with the option *varIdent*. Standardized residuals were plotted against fitted data and no evidence for heterogeneity was found [62]. Distribution of residuals was considered normal if absolute skewness was below 2 and excess kurtosis below 4 [63]. The influence of outlier residuals that fell more than 3 standard deviations from the mean was assessed by comparing models with and without outlier data points; all models were qualitatively equivalent, so no data points were excluded. We evaluated the significance of fixed effects and their interaction by comparing models using log-likelihood ratio tests following deletion of terms. Additionally, for models with a significant interaction between treatment and time, pairwise comparisons between immune-

challenged and control-injected ants at each time post-injection were conducted (function *emmeans* from package *emmeans*).

To analyse CHC profiles, the area under a peak from the GC was integrated to calculate the relative proportions of the different compounds. Three peaks contained two compounds of similar retention times and had to be pooled for analysis. A permutational multivariate analysis of variance using Bray–Curtis Distance Matrices (ADONIS) was performed to analyse the effects of the treatment (immune-challenged versus control-injected versus naive), the social environment (alone versus in group) and their interaction on CHC profiles. Pairwise comparisons (with the Benjamini–Hochberg adjustment for multiple testing) were then performed between treatments for ants that were kept alone. A discriminant analysis of principal components was used to visualize differences in CHC profiles between experimental groups. To identify the individual compounds that differed between groups, a random forest analysis was performed [64].

## 3. Results

### (a) Pathogen exposure
#### (i) Survival assay

Pathogen exposure interacted with the social environment to shape survival (Cox proportional hazard model; interaction: $\chi^2_1 = 20.70$, $p = 5.355 \times 10^{-6}$; electronic supplementary material, figure S1). Overall, 58.3% (70 out of 120) of the pathogen-exposed ants died within 13 days, all of them of confirmed *Metarhizium* infection. Pathogen-exposed ants died significantly faster than sham-exposed ants (pairwise log-rank tests: pathogen-exposed alone versus sham-exposed alone, $p < 2.000 \times 10^{-16}$; pathogen-exposed alone versus sham-exposed grouped, $p < 2.000 \times 10^{-16}$; pathogen-exposed grouped versus sham-exposed alone, $p = 0.007$; pathogen-exposed grouped versus sham-exposed grouped, $p = 0.019$). Moreover, all pathogen-exposed ants kept alone died between days 3 and 9, significantly faster than pathogen-exposed ants kept in groups ($p < 2.000 \times 10^{-16}$). Thus, the presence of nest-mates dramatically increased the survival of pathogen-exposed ants, but not of sham-exposed ants (sham-exposed alone versus sham-exposed grouped, $p = 0.560$).

#### (ii) Behavioural assay

The effect of pathogen exposure on received grooming varied over time (GLMM; interaction: $\chi^2_{16} = 67.07$, $p = 3.233 \times 10^{-8}$; electronic supplementary material, figure S2): pathogen exposure elicited a strong behavioural response in nest-mates, with pathogen-exposed ants receiving more grooming than sham-exposed ants immediately after treatment (mean and 95% confidence intervals, 0 h: pathogen-exposed: 530 s (483–567), sham-exposed: 92 s (50–142), $p < 0.001$; electronic supplementary material, table S1) and 12 h post-exposure (pathogen-exposed: 106 s (36–186), sham-exposed: 7 s (0–19), $p < 0.001$). Thus, ants responded to fungal exposure by increasing allogrooming, which probably caused the increase in survival of pathogen-exposed ants kept in groups.

### (b) Immune challenges
#### (i) Behavioural assays

Immune challenges increased allogrooming in a time-dependent manner (GLMM; interaction: $\chi^2_8 = 15.93$, $p = 0.043$; figure 1). Following an early increase in allogrooming in both treatments (6 h post-injection), immune-challenged ants

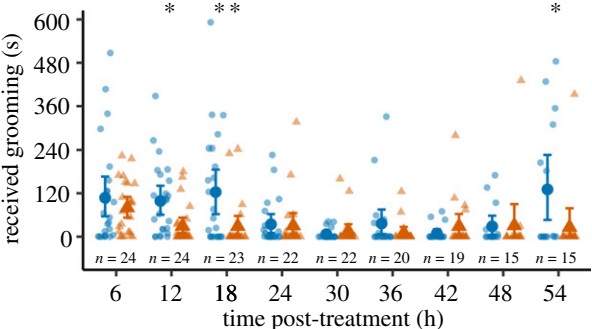

**Figure 1.** Grooming received by immune-challenged (blue) and control-injected (orange) ants over 10 min. Large symbols are mean ± bootstrapped 95% confidence intervals. Small symbols and sample sizes indicate replicate colonies. $**p < 0.01$, $*p < 0.05$.

received more grooming than control-injected ants early in the experiment (mean and 95% confidence intervals: 12 h: immune-challenged, I: 99 s (60–142), control-injected, C: 30 s (10–53), $p = 0.027$; 18 h: I: 123 s (64–185), C: 29 s (5–59), $p = 0.010$; electronic supplementary material, table S2), and again 54 h post-injection (I: 131 s (47–222), C: 26 s (0–79), $p = 0.019$). Thus, the immune status of the challenged—but uninfected—ants induced a qualitatively similar behavioural response (allogrooming) as exposure to live pathogens in their nest-mates (electronic supplementary material, figure S2).

Immune challenges not only elicited a behavioural response in nest-mates but also affected the behaviour of the challenged ants. Treatment and time interacted to shape all three observed behaviours (LMM, isolation: $\chi^2_8 = 18.16$, $p = 0.020$, figure 2a; locomotor activity: $\chi^2_8 = 21.28$, $p = 0.006$, figure 2b; walking speed: $\chi^2_8 = 15.86$, $p = 0.043$, figure 2c). Isolation was higher in control-injected ants than in immune-challenged ants early in the experiment (15–21 h: I: 55% of time (43–68), C: 72% (60–78), $p = 0.015$; 21–27 h: I: 61% (48–73), C: 73% (60–83), $p = 0.032$; electronic supplementary material, table S3) and towards the end of the experiment (45–51 h: I: 48% of time (32–64), C: 63% (49–74), $p = 0.046$; 51–54 h: I: 50% (38–61), C: 72% (67–77), $p = 0.002$). Similarly, control-injected ants showed higher locomotor activity than immune-challenged ants early in the experiment (9–15 h: I: 10% of time (7–15), C: 25% (15–37), $p = 0.001$; 15–21 h: I: 13% (7–22), C: 24% (15–34), $p = 0.004$) and to a lesser extent towards the end of the experiment (39–45 h: I: 9% of time (5–13), C: 17% (10–24), $p = 0.031$; 45–51 h: I: 8% (6–11), C: 17% (10–25), $p = 0.013$). Finally, immune-challenged ants had slower walking speed than control-injected ants early (9–15 h: I: 1.02 mm s$^{-1}$ (0.82–1.24), C: 1.35 mm s$^{-1}$ (1.08–1.69), $p = 0.006$) and to a smaller degree later in the experiment (39–45 h: I: 0.77 mm s$^{-1}$ (0.63–0.93), C: 0.97 mm s$^{-1}$ (0.76–1.20), $p = 0.037$). Differences in isolation, activity and walking speed between immune-challenged and control-injected ants thus followed a temporal dynamic comparable, but not identical, to that observed for allogrooming (figure 1), dominated by early, transient behavioural effects. Visual inspection of ant trajectories (figure 2d) indicated that while the activity levels of immune-challenged ants were reduced, their spatial distribution showed considerable overlap with that of other ants and was centred on the brood, suggesting no social exclusion or nest avoidance.

This was confirmed in network analyses. Contact networks generated by automated tracking (figure 2e) accurately reflected manually scored contact networks (Pearson correlations: $r_{34}$: 0.84–0.99, all $p < 1.333 \times 10^{-10}$). Both node eigenvector centrality and strength were higher in immune-challenged ants than in control-injected ants (LMM, eigenvector centrality: $\chi^2_1 = 8.25$, $p = 0.004$, figure 2f; strength: $\chi^2_1 = 5.36$, $p = 0.021$, figure 2g; electronic supplementary material, table S3) and decreased over time (eigenvector centrality: $\chi^2_8 = 34.51$, $p = 3.279 \times 10^{-5}$; strength: $\chi^2_8 = 164.24$, $p < 2.000 \times 10^{-16}$). This temporal dynamic was also reflected in a decrease in colony-level network strength over the course of the experiment (LMM, time: $\chi^2_8 = 331.84$, $p < 2.000 \times 10^{-16}$; electronic supplementary material, figure S3). Finally, skewness of contacts increased over time ($\chi^2_8 = 48.40$, $p = 8.270 \times 10^{-8}$; electronic supplementary material, figure S4) but was not affected by the treatment ($\chi^2_1 = 0.78$, $p = 0.376$). Thus, immune-challenged individuals occupied a more central network position but did not interact with a reduced set of individuals compared to control-injected ants. Interestingly, changes in behaviour measured by automated tracking were sometimes apparent throughout the experiment (figure 2f,g) and not limited to times when a transient increase in grooming was observed (figure 1). This suggests that immune challenges increase contacts other than grooming and/or that automated tracking is more sensitive than manual tracking to small increases in grooming. Automated analyses also revealed that control-injected ants often had behavioural phenotypes intermediate between naive and immune-challenged ants (figure 2b,c,f), potentially because injection-induced wounding itself induced an immune response in control-injected ants.

### (ii) Cuticular hydrocarbon profiles

GC-MS analyses revealed a total of 18 unique hydrocarbons on the cuticle (electronic supplementary material, table S4). Both the social environment (ADONIS: d.f. = 1, $R^2 = 0.07$, $p = 0.030$; electronic supplementary material, figure S5) and the treatment (d.f. = 2, $R^2 = 0.12$, $p = 0.018$), but not their interaction (d.f. = 2, $R^2 = 0.02$, $p = 0.630$), affected CHC profiles. However, the effect of the treatment was driven by differences between naive and immune-challenged ants kept alone ($R^2 = 0.26$, $p = 0.021$), while control-injected ants did not differ from either naive ($R^2 = 0.12$, $p = 0.207$) or immune-challenged ants ($R^2 = 0.09$, $p = 0.207$), indicating that immune challenges *per se* did not affect CHCs profiles. A random forest analysis identified methyl branched alkanes as the main explanatory compounds for group identity between treatments, in particular for the social environment (electronic supplementary material, figures S6–S8).

## 4. Discussion

We combined pathogen exposure and immune challenges with manual and automated behavioural analyses to quantify how ant colonies respond to perceived infection risk. We consistently observed increased social interactions with individuals that were perceived to be infectious. In both pathogen-exposed and immune-challenged ants, this manifested as an increase in grooming received from nest-mates. Additionally, colony-wide automated tracking revealed that immune-challenged ants generally spent more time in

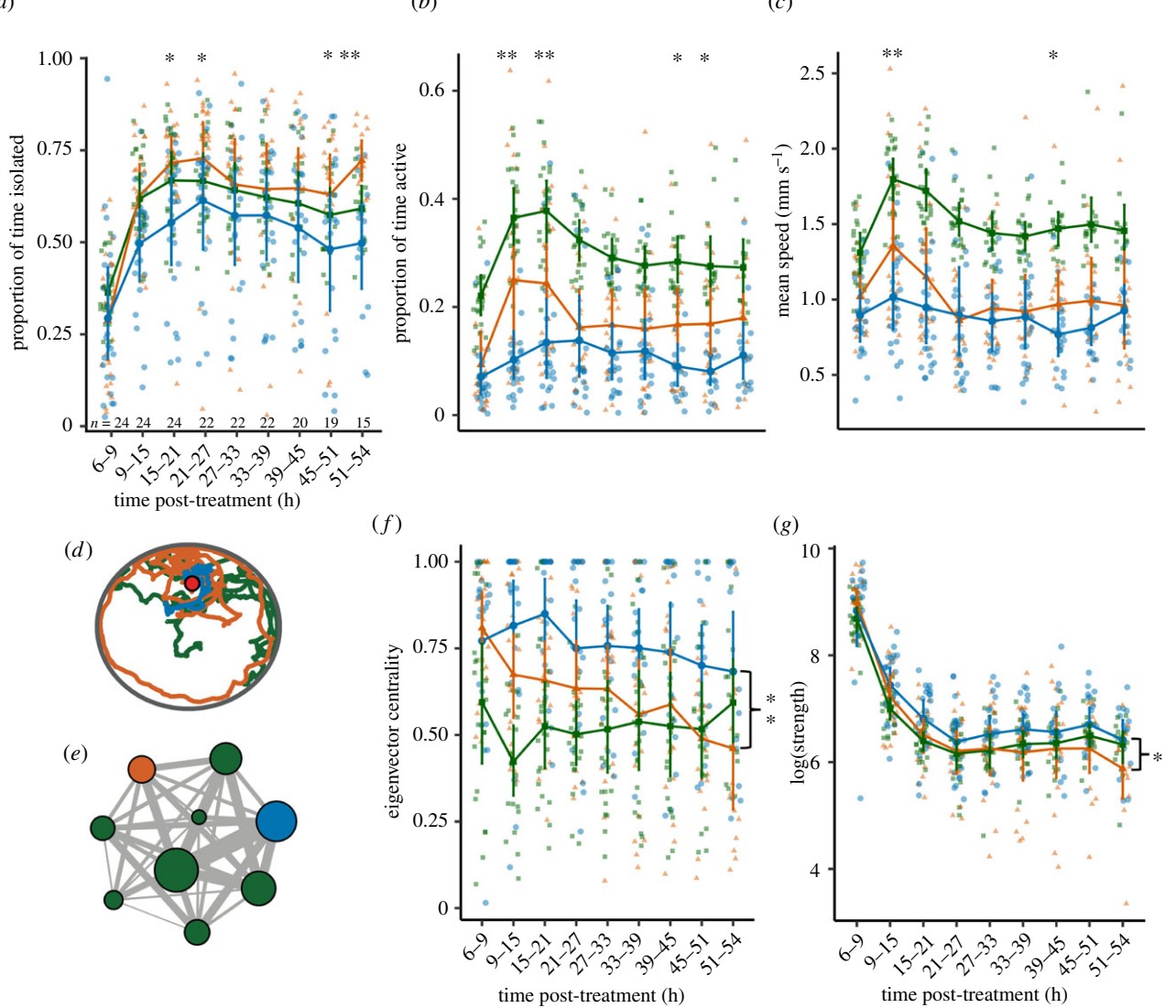

**Figure 2.** Behaviour of immune-challenged (blue), control-injected (orange) and naive ants (green, *a–c*, *g*: average of all naive ants in each colony, *f*: one random naive ant per colony) over time. (*a–c,f,g*) Large symbols represent mean ± bootstrapped 95% confidence intervals. Small symbols indicate replicate colonies. Sample sizes for each time point are shown in (*a*). (*d*) Example trajectories of three ants over 10 min. The red dot indicates the position of the brood. (*e*) Example contact network based on 6 h of tracking. The node size is proportional to the eigenvector centrality. The edge width is proportional to the contact duration. **$p < 0.01$, *$p < 0.05$. Statistical results refer to comparisons between immune-challenged and control-injected ants.

physical contact (irrespective of the exact type of interaction) with their nest-mates and occupied a more central network position than control ants. Because node centrality increases the potential for an individual to spread disease [59], the observed behaviour would be predicted to increase, not decrease, the transmission of actual pathogens. Furthermore, this predicted risk of increased social contacts was not counteracted by skewing social contacts towards a reduced set of social partners. Taken together, these results indicate a lack of avoidance behaviour towards either pathogen-exposed or immune-challenged individuals in favour of a general 'caring' strategy towards social partners that are infectious or perceived to be infectious, which we discuss in the light of the potential cost–benefit structure of expressing care under different scenarios.

Our work adds to a mounting body of evidence showing that group-living animals, from insects [31,43] to mammals [30,65–67], can detect the immune status of their social partners. However, the nature of the behavioural responses displayed towards immune-challenged individuals following their detection varies across contexts [30,68] and systems

[15,33,43,44,67], ranging from strategies that increase social contact (e.g. allogrooming [16,17,43,69]) to strategies that decrease them (e.g. social exclusion [15,21,65]). This qualitative variation is also observed towards pathogen-exposed individuals, where it is thought to reflect the costs and benefits associated with caring for infectious social partners. For example, termites shift from grooming to cannibalizing infected nest-mates depending on infection stage and thus, the likelihood that the benefits of grooming outweigh its costs [70]. Similarly, ants perform risk-adjusted care by reducing grooming towards nest-mates carrying a pathogen that they are more susceptible to [71]. While the expression of allogrooming is adjusted to its risks, these risks are not necessarily high. In the case of fungal spores, for example, grooming rarely leads to secondary infections [17,18,71]. In fact, exposure to sublethal doses of spores via grooming can even induce immune responses that protect groomers against subsequent exposure to the same pathogen [19]. Here, grooming had clear benefits for pathogen-exposed individuals (it increased their survival), but no obvious cost for the rest of the colony over the course of the experiment.

Further work will determine whether this strategy plastically changes if the costs of allogrooming are increased (e.g. with more infectious pathogens) or its benefits decreased (e.g. with advanced infections, the outcome of which cannot be improved by allogrooming).

While the costs and benefits of grooming nest-mates exposed to fungal spores are well characterized, the potential benefits of grooming nest-mates with elevated immune activity are less immediately clear. Here, we used an immune elicitor simulating a fungal infection, and cannot rule out that cues tied to immune responses to this particular elicitor were sufficient to induce a behavioural response (grooming) specifically targeted at fungal pathogens that infect through the cuticle. However, other immune elicitors [33,43], and wounding [72], have been reported to induce grooming in social insects, showing that grooming is not a behavioural response specific to fungal pathogens. More plausibly, immune challenges produce sickness cues that are not pathogen-specific, and allogrooming is expressed as general care-taking behaviour. Indeed, allogrooming can benefit the receiver via mechanisms other than the physical removal of infectious particles, for example, by spreading antimicrobial compounds onto the cuticle [73].

Our approach combining manual and automated behavioural analyses following immune challenges shows a largely congruent general pattern (increases in social contacts), but important differences (duration of effects), which highlights the advantages and challenges associated with each type of analysis. Manual approaches allow us to quantify specific behaviours, but the amount of data that can be analysed this way is necessarily limited. By contrast, automated approaches allow us to process much larger amounts of data (e.g. from more individuals, over longer time periods), and can improve analyses quantitatively (e.g. by increasing the likelihood to detect small effects) and qualitatively (e.g. by allowing to analyse group-level behaviour). However, the majority of automated tracking studies in social insects [14,54,74], including ours, define interactions based on spatial location, which does not differentiate different types of interactions (e.g. grooming, aggression, trophallaxis) that may have different—and possibly opposite—impacts on disease dynamics. Overcoming this limitation will be facilitated by the ongoing development of techniques enabling the automated detection of specific social behaviours (e.g. trophallaxis [75,76]) and the study of their role in promoting or preventing disease spread [69].

In contrast with reports from honeybees [31,33] and ant brood [12], we found no evidence that CHCs mediate the observed behavioural responses, which may thus rely on other chemical cues (e.g. volatile compounds) [77] or on behavioural cues [78]. Instead, we found an effect of the social environment (alone versus in groups), indicating that social interactions affect CHC profiles [48,49]. This highlights that the causal relationship between behaviour and CHCs can go both ways: not only can changes in CHC profiles affect social behaviour but social interactions can affect CHC profiles. Thus, drawing conclusions on the chemical cues driving behavioural changes requires accounting for effects of social interactions themselves.

Social immunity has so far primarily been studied in eusocial insects where a strict reproductive division of labour between fertile queens and largely sterile workers creates asymmetries in value across colony members [7,27,79]. Here, we show that an ant with maximal relatedness but minimal reproductive division of labour and, therefore, homogeneous value across individuals displays disease-relevant behaviour (survival-enhancing allogrooming) that is qualitatively similar to the forms of social immunity found in 'standard' social insects [7,79]. This and other species with unconventional social structures can provide insight into the generality of collective defences against disease across social insects, and perhaps across group-living species, by allowing us to explore a larger part of the social 'parameter space'.

Data accessibility. Data and code for this article are available from the Dryad Digital Repository: https://doi.org/10.5061/dryad.wwpzgmshb [80].

Authors' contributions. G.A.: data curation, formal analysis, investigation, methodology, project administration, software, supervision, validation, visualization, writing—original draft; L.V.U.: conceptualization, data curation, formal analysis, funding acquisition, investigation, methodology, writing—review and editing; E.F.: conceptualization, data curation, formal analysis, methodology, visualization, writing—review and editing; J.B.: investigation; A.G.: formal analysis, methodology, software, writing—review and editing; T.S.: methodology, resources, writing—review and editing; D.J.C.K.: funding acquisition, resources, supervision, writing—review and editing; Y.U.: conceptualization, funding acquisition, investigation, methodology, project administration, resources, supervision, writing—original draft, writing—review and editing. All authors gave final approval for publication and agreed to be held accountable for the work performed therein.

Competing interests. The authors declare no competing interests.

Funding. Open access funding provided by the Max Planck Society.

This work was supported by the Swiss National Science Foundation (grant no. PZ00P3_168066) and the European Research Council (ERC) under the European Union's Horizon 2020 research and innovation program (grant agreement no. 851523) to Y.U.; Independent Research Fund Denmark Sapere Aude: DFF-Research Talent grant no. 4090-00032 to L.V.U.; Human Frontiers Science Program long-term fellowship LT001049/2015 to A.G.; a Pew Biomedical Scholar Award to D.J.C.K.; a Swiss National Science Foundation and Advanced European Research Council grant to Laurent Keller.

Acknowledgements. We thank the Center for Advanced Modelling of Science and the Division de Calcul et Soutien à la Recherche of the University of Lausanne for help with the computational approaches and access to the computing infrastructure; Simon Garnier, Sylvia Cremer, Michel Chapuisat, Tom Kay and Laurent Keller for helpful feedback. This is Clonal Raider Ant Project paper no. 18.

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
