## [Peer Review File · Proceedings of the Royal Society B: Biological Sciences]

Review History

RSPB-2020-2570.R0 (Original submission)

Review form: Reviewer 1

Recommendation

Major revision is needed (please make suggestions in comments)

Scientific importance: Is the manuscript an original and important contribution to its field?

Excellent

General interest: Is the paper of sufficient general interest?

Excellent

Quality of the paper: Is the overall quality of the paper suitable?

Excellent

Is the length of the paper justified?

Yes

Should the paper be seen by a specialist statistical reviewer?

No

Do you have any concerns about statistical analyses in this paper? If so, please specify them explicitly in your report.

Yes

It is a condition of publication that authors make their supporting data, code and materials available - either as supplementary material or hosted in an external repository. Please rate, if applicable, the supporting data on the following criteria.

Is it accessible?

Yes

Is it clear?

Yes

Is it adequate?

Yes

Do you have any ethical concerns with this paper?

No

Comments to the Author

Comments:

Overall, this is an interesting and timely study combining observations of behaviors and automated tracking to explore how queenless ants respond to fungus exposed/immune-challenged conspecifics. The main strength of the manuscript lays (I) in the multiple methods to expose/immune-challenge ants, (II) the comparative nature to previous studies in ants that focus on species with division of labor, and (III) the fact that group-wide observations were collected through automated movement tracking after the challenge experiments. I do, however, have concerns about certain interpretations of the manuscript. First, detection of infection risk is well described in other study systems, and I think the main emphasis should be more on the fact that the authors collect group-level data. Second, the authors repeatedly discuss the results of tracking and behavioral observations of allogrooming interchangeably even though spatial associations and physical interactions should not be considered the same. Third, the differentiation between ants changing their behavior towards “exposed” and “immune-challenged” conspecifics needs to be clearer. Finally, I think the cost-benefit structure of caregiving could be explored in a bit more detail for both, fungal exposure and immune-challenges. Given these concerns, I would recommend major revisions before publishing the study. See below my specific comments.

Title:

I realize this is a personal preference, but I prefer descriptive titles that highlight the main result in more detail (e.g. in what way is network position affected?). “Immune status” also suggests to me that the immune status was measured instead of experimentally changed. This could be clarified by using “immune-challenged”.

Introduction:

General comments:

The introduction is well written, but I disagree slightly with the main points of the manuscript. First, detection of infection risk is well described in several species including humans (most prominently in the expression of avoidance responses). Experiments often focus on focal individuals or pairs of animals (I suggest adding some citations, see below). I think what makes this study somewhat novel is the fact that it integrates responses of multiple individuals simultaneously and focuses on their entire network. The authors mention this in lines 76-78 but I suggest emphasizing this more.

Second, the authors mainly emphasize the division of labor as a prominent driver of social immunity in their introduction, but also discussion. While I agree with the authors that one of the drivers of collective defenses in social insects seems to be division of labor and asymmetry in “importance” of individuals, high relatedness between individuals seems at least equally important. The authors state that pathogen exposure is potentially more costly for a clonal raider ant than for a sterile worker but if that clonal raider ant assists a cohort member that is genetically nearly identical, wouldn’t these costs become negligible? To me, one novelty of the study lays in the fact that it is able to differentiate the two “conditions” of social immunity in social insects, (i) protecting more valuable individuals (such as the queen) and (ii) high relatedness, and that these behaviors are observed despite one is missing (division of labor). This could be clarified a bit more in the introduction (and discussion).

Minor comments:

Lines 39-40: I have not seen the term “social immunity” used broadly outside social insects where it describes a collective group defense (and includes seemingly altruistic behaviors) against pathogens that benefits the whole colony. I suggest either changing “many social species” to “many social insect species” or citing examples where social immunity and collective group defense is used in systems outside social insects.

Lines 46-47: Isn’t there also the possibility of the spread of sublethal doses and hence, immunization within the colonies? This could be mentioned here, or somewhere else in the manuscript specifically in terms of the alleviated costs of continued allogrooming of exposed conspecifics.

Lines 63-64: Detection of infection risk has been studied in, for instance, mice, guppies, and mandrills, amongst others. In humans, this seems to be driven by a generalized “disgust-eliciting” response and a recent review outlines some of the key research nicely (Townsend et al, 2020, see below). The authors also mention studies in honeybees and ant pupae suggesting that detection of infection risk has been studied in social insects as well. I, therefore, do not agree with the statement that detection of infection risk is poorly understood. To clarify this, I would suggest adding more citations to this section and emphasize the fact that the presented study focuses on whole-network responses rather than just detection itself.

Suggested citations:

Boillat et al 2008; The vomeronasal system mediates sick conspecific avoidance, *Current Biology*.

Stephenson et al. 2018; Transmission risk predicts avoidance of infected conspecifics in Trinidadian guppies. *Journal of Animal Ecology*.

Poirotte et al 2017; Mandrills use olfaction to socially avoid parasitized conspecifics. *Science Advances*.

Townsend et al 2020; Emerging infectious disease and the challenges of social distancing in human and non-human animals. *Proceedings B*.

Lines 67-70: As I understand this refers to a generalized disgust response. This has been discussed in a recent review by Townsend et. al. (see previous comment) which could be included as a citation. I would also suggest defining “slower” in this context. To me, if individuals can recognize general cues of infection (instead of relying on pathogen-specific cues), this could lead to a more rapid response.

Lines 82-85: For the broad readership of the target journal I suggest describing in a bit more detail how reproduction in this ant species works. How are these cohorts produced? Do individual ants clone themselves?

Lines 88-89: Please clarify minimal reproductive division of labor. Is there reproductive division of labor in this species to some extent?

Material and Methods:

Lines 113-120: I would suggest describing the infection-transmission dynamics of *M. robertsii*. It would be important to explain the difference between exposure and infection in this system (when does the fungus infect the ant? Is there a lag time between exposure and infection/physiological response? When do conspecifics respond to exposure versus infection on this timeline?).

Line 115: I suggest clarifying the term "germination check" for a broad readership.

Lines 124-125: Is there a reason why ants of different ages were used? Please explain.

Lines 136-138: How was determined whether age had a significant effect or not? This should be mentioned briefly.

Lines 139-140: Minor comment. At this point, I had to go back to check for the fixed factor descriptions above. I wonder if it would be helpful to include the actual factors in brackets such as treatment (infected/uninfected), social environment (alone/grouped)? The authors use a similar way of describing factors in their model in lines 146-148.

Line 144-145: Why is grooming received in minutes treated as a binomial response variable as opposed to a continuous response in a linear mixed effect model? Please clarify.

Line 146: I am confused about grooming received and the rounding of numbers. For instance, if an ant received 1.5 minutes in grooming, was this rounded up to 2 minutes? Why was the response variable rounded up to the next integer?

Lines 155-156: Was the volume of 0.1 ul used based on other studies? I also assume that, because these ants seem to be similar in morphology (and probably weight), there were no differences in the actual dosage each ant received.

Lines 156-159: Was survival of immune-challenged ants determined similarly to the fungal exposures above?

Lines 161-162: I noticed that ants for this assay were from genotype B and ants in the infection assay above were from genotype A. Are there significant differences between those genotypes or are they similar, please clarify this briefly at some point in the methods?

Lines 168-170: Defining social network edges requires deciding what minimum value of proximity or duration is sufficient for an association event. The authors define an isolation distance of >1mm (half a body length) and interaction distance of (< 1mm) and I appreciate the validation of the approach by observing the ants. A recent article observed effect sizes across a wide range of how an association is defined by re-sampling the collected data (Ripperger et al 2020, Behavioral Ecology). It would be fascinating to look at similar effects here, i.e. at which definition of an association does the effect disappear and what types of spatial associations are more or less important (with an emphasis on how pathogens spread, from close contact to further distance). I realize that this is beyond the scope of this study but might represent an interesting avenue for future research.

See for details:

Ripperger et al., 2020, Tracking sickness effects on social encounters via continuous proximity sensing in wild vampire bats. *Behavioral Ecology*.

Lines 174: I urge the authors to differentiate physical interactions (such as grooming) from spatial associations (distance to each other) more carefully. Being near to one another does not necessarily show that these ants groom each other.

Lines 208-213: As far as I understand here, the network characteristics (eigenvector centrality, strength, and skewness) are based on non-independent data. Please clarify if, and how you are dealing with non-independence in your analysis? Alternative approaches include node-label network permutations (see Ripperger et al, 2020, *Behavioral Ecology*).

Results:

Lines 253- 257: Instead of the mean difference and SEs, the authors could mention the means and bootstrapped confidence intervals for both treatment groups 0, 12, 18, and 48 hours post-exposure (and for other results throughout). I also recommend summarizing the means and bootstrapped confidence intervals shown in each plot in supplementary tables. I would suggest outlining the method of bootstrapping CIs at some point either in the results or the material and methods section (e.g. how often was the data resampled?).

Figure 2 (Lines 267-269): Do the authors have an observation for time = 0 (or shortly after the challenge) that could be added? If possible, I think this would be interesting to show because it would suggest more clearly that the physiological response of immune-challenged ants (delayed for about 9-15 hours, see Figure 3) increases the grooming response of conspecifics.

Discussion:

General comments:

>One of the interesting findings of the paper is that ants increase grooming of fungus-exposed conspecifics which could increase the exposed individual's survival. In this case, the cost-benefit structures seem to be less ambiguous. The exposed individual has a higher chance of survival (because of the removal of spores) and the groomer, even though at greater risk of infection, might even gain some personal and colony-level benefits through exposure-immunization as shown in other ant species. I would like to see more discussion about the finding that allogrooming towards immune-challenged ants also seems to increase (even though it might not increase their survival). How can this potentially benefit the immune-challenged ant and the ant that performs the grooming? Does this suggest that this care-taking behavior is a generalized response and less dependent on the identity of the pathogen? What is the difference between exposure and "immune-challenged" in the cost-benefit balance for both the groomer and the receiver of grooming?

>The authors mainly highlight reproductive division of labor as an important difference between other social insect systems and their own. While I do believe this to be an important differentiation, I also think that the fact that all individuals are highly related to each other needs to be mentioned and discussed more explicitly. An individual might gain a lot from helping a nearly clonal conspecific in terms of its indirect fitness, especially if costs to itself are minor.

>While the authors mention their results of the automated tracking (Lines 359-363) I think it is important to differentiate here, and throughout the manuscript (see also methods part), that physical closeness (<1mm) does not equal physical interactions (i.e. grooming). In fact, networks constructed by using physical interactions such as grooming and spatial associations can differ quite a bit. For instance, an individual may be near a lot of others but does not receive grooming. Importantly, both can have profound effects on pathogen transmission which depends on the transmission pathway a pathogen takes. If a pathogen is transmitted through physical interactions (i.e. through grooming) a network constructed based on spatial associations might not predict its spread accurately. This could be discussed specifically.

Minor comments

Line 335-337: Do the authors have specific experiments in mind that could give us insights into the generality of collective defenses against diseases across group-living species? For instance, is it possible to vary kin relationship structure to some extent in these experimental colonies?

Lines 341-343: As I understand from the fungal pathogens used, an ant can be exposed (and infectious) but not infected yet (i.e. the fungus has not entered the body yet). It seems to me that ants could recognize exposure before the ant is infected (i.e. immune-challenged) and increase grooming. Could this be by detecting cues associated with the pathogen itself on the cuticle surface? In the immune-challenge experiment, however, recognition might be driven by behavioral/physiological changes related to the physiological response. This difference, if applicable, should be highlighted more explicitly.

Lines 348-350: The statement “the causal relationship between behavior and CHC profiles can go both ways” is confusing. Please revise this paragraph.

Lines 366-369: The authors mention that future work could determine whether this strategy plastically changes based on, for instance, risk adjustment of groomers. As I understand there are studies that show risk adjusted grooming in ants and I suggest citing them here.

Suggested citation:

Konrad et al 2018, Ants avoid superinfections by performing risk-adjusted sanitary care, PNAS.

Review form: Reviewer 2

Recommendation

Reject – article is not of sufficient interest (we will consider a transfer to another journal)

Scientific importance: Is the manuscript an original and important contribution to its field?

Marginal

General interest: Is the paper of sufficient general interest?

Poor

Quality of the paper: Is the overall quality of the paper suitable?

Acceptable

Is the length of the paper justified?

Yes

Should the paper be seen by a specialist statistical reviewer?

No

Do you have any concerns about statistical analyses in this paper? If so, please specify them explicitly in your report.

No

It is a condition of publication that authors make their supporting data, code and materials available - either as supplementary material or hosted in an external repository. Please rate, if applicable, the supporting data on the following criteria.

Is it accessible?

Yes

Is it clear?

Yes

Is it adequate?

Yes

Do you have any ethical concerns with this paper?

No

Comments to the Author

In this study, the authors investigated the expression of social immunity in a queenless ant species. First, they show that workers survive better to infection with pathogens in presence of other workers compared to alone and that workers exposed to a pathogen received more allogrooming compared to non-infected workers. Then using both visual observations and an automatic tracking system, they show that immune-challenged workers are more often isolated in the nest, tend to be less active, as well as occupy a more central network position compared to control-challenged workers.

I love this topic and this system, and I was therefore very eager to read this manuscript. Unfortunately, it mostly presents results that are already well known in other species, based on methods that have also been used earlier to test the same questions. So, I do not really understand the novelty of the presented work, and to what extent it should be of interest to a broad readership. In particular, it is unclear as to why clonal systems (which is the novel aspect of the work) should have completely different forms of social immunity (or no social immunity at all). What are the predictions in terms of the proposed measurements (do less allogrooming? Be more active? Occupy different positions in the network?)? My take was that social immunity should be present there and in a not so much different form compared to the other ant systems... And this is exactly what you show. I do understand that it is generally worth having a look at many species to better understand complex processes such as social immunity, but I don't think that the associated results (if they do not emphasize major differences or induce major changes in the theory) can be of interest to the broad readership of Proc B. This is even clearer with the discussion, which is extremely short and does not put the results in a broad context. At the end of the reading, it is actually difficult to understand how the presented results provide novel and interesting insights into our general understanding of the evolution and expression of social immunity, or on how social species deal with infected group members. On other words, the discussion illustrates that this is a very solid but also very, very specific study that is mostly addressed to researchers looking for more taxonomically diverse examples of a system that is already well known. Overall, I strongly recommend the authors to rewrite their introduction to better emphasize the novelty of their work and provide clear rationales for the study, to (really) meat up their discussion to help readers that are not working on clonal ants to put the results into broad perspectives and then to submit their (solid!) work to a more specialised journal.

Below, you can find a few other comments:

Abstract: the abstract is oddly written, as it is difficult to understand what is the main question (and why it is important). For instance, it seems that the main question is (L25-26) "how this is achieved remains unclear", which is clearly not the main focus of the study.

L62: I think that this study (see below) could be cited here, as it provides support to this claim. Stroeymeyt, N., Grasse, A. V., Crespi, A., Mersch, D.P., Cremer, S., Keller, L., 2018. Social network plasticity decreases disease transmission in a eusocial insect. *Science* 362, 941-945. <https://doi.org/10.1126/science.aat4793>

L63-70: This paragraph comes a bit out of the blue and lacks references. However, paragraph L100-109 contains all the necessary references. I suggest combining these two parts into a single paragraph.

L78: I have the feeling that it is an overstatement, as at least one other study do this. (Stroeymeyt, N., Grasse, A. V., Crespi, A., Mersch, D.P., Cremer, S., Keller, L., 2018. Social network plasticity decreases disease transmission in a eusocial insect. *Science* 362, 941–945).

L111: The material and methods part is difficult to read, as it is particularly succinct and lacks any general description. For instance, what "pathogen exposure" stands for, as it contains survival assays (to test what?) and behavioural assays. This is the same for the "immune-challenges" part. Please, rewrite this entire part to facilitate its reading (first explain the general idea, and then provide the details).

L161: Is there any evidence that you can compare patterns obtained in genotype A (first part of the study) with patterns obtained in genotype B (this part) and with patterns obtained in genotype C (part 3 on CHCs)? Why did you use each genotype for each sperate experiment?

L243: I am afraid that the reported effect is not a clear demonstration of social immunity, as it may simply reflect that the stress of social isolation specifically hampers the immune response of isolated workers. This alternative interpretation receives support in gregarious earwigs: Kohlmeier, P., Holländer, K., Meunier, J., 2016. Survival after pathogen exposure in group-living insects: don't forget the stress of social isolation! *J. Evol. Biol.* 29, 1867–1872.
<https://doi.org/10.1111/jeb.12916>

Review form: Reviewer 3

Recommendation

Accept with minor revision (please list in comments)

Scientific importance: Is the manuscript an original and important contribution to its field?

Good

General interest: Is the paper of sufficient general interest?

Good

Quality of the paper: Is the overall quality of the paper suitable?

Good

Is the length of the paper justified?

Yes

Should the paper be seen by a specialist statistical reviewer?

No

Do you have any concerns about statistical analyses in this paper? If so, please specify them explicitly in your report.

No

It is a condition of publication that authors make their supporting data, code and materials available - either as supplementary material or hosted in an external repository. Please rate, if applicable, the supporting data on the following criteria.

Is it accessible?

N/A

Is it clear?

N/A

Is it adequate?

N/A

Do you have any ethical concerns with this paper?

No

Comments to the Author

In this paper, the author nicely combine manual and automated behavioural analyses to test whether ants that are challenged by a non pathogenic elicitor change their behaviour in a way that could affect social immunity such as allogrooming, activity, network position in the nest. They used a very interesting model system: the parthenogenetic and queenless clonal raider ant, *Ooceraea biroi*. This species offers the advantage to have workers genetically and morphologically homogeneous, that are totipotent but with no reproductive conflicts because of parthenogenetic reproduction. As a first step, the authors show that this peculiar species also displays social immunity as observed in other more classical ant species with division of labour. Ants exposed to pathogens (the classically used *Metarhizium* fungus) have a lower survival when they are kept alone than in small groups. They also show that pathogens exposed ants have higher rate of allogrooming. As a second step, they challenged a worker in a group of 9 ants and 5 larvae by a known elicitor of the insect immune system (*Saccharomyces cerevisiae* cell walls) that have no pathogenic effect, and compared its behaviours to a control worker of the same group (injected with just the PBS). They combine manual and automated behaviour tracking to compare allogrooming, activity, walking speed, isolation and network position. Immune challenged workers received more allogrooming, have a reduced activity level and have a more central network position and were less isolated than the control workers. The paper is nicely written, the analysis properly conducted.

Comments:

M&M

Line 129-130: what means old and young in terms of day old? How were the age of the ants known?

Genotypes: the genotypes used were different for each experiment (A for pathogene exposure , B for immune challenge and D for the analysis of CHC profil). Are there some specific reasons for using different genotypes? Do their lineages differ by some life history traits or behaviours that could affect the results?

Line 136-138: how was the effect of age tested? This should be said and the statistical results given to confirm this assertion (in a supplementary material if needed).

Line 145-146: not clear to me why the response variable "received grooming" should follow a binomial distribution. For a binomial model, the response variable should be either 0/1 or a number of responses over a number of trials. I guess that received grooming belong to the second type of response variable. In that case, the risk of over dispersion is important and is known to potentially affect the statistical significance of the factors if not taken into account. But may be I did not understand properly how was built the binomial model.

Line 156-158: in the immune challenge experiment, workers in the control-injected treatment also have their immune defence challenged as injection with PBS, by piercing the membrane can induce an immune response. This does not affect the conclusion of the paper, but just if the author would have also considered in their comparison a focal naïve worker, the effect of immune challenge could even have been more pronounced. Given that all workers were video tracked, I am wondering why the authors did not choose to also include a focal naïve worker in

each group. This is particularly surprising given that for the study of CHC, the authors considered the three groups (immune challenged, control injected and naïve workers).

Line 160-162: did the authors have some experimental evidence that the behaviours of the ants in such small experimental groups of 9 ants and 5 larvae are representative of the behaviours display in full colonies? What is the mean size of colonies?

Results:

Fig1 and Fig2: I realised reading these two figures that the time post-treatment was not similar in the pathogen exposure experiment and the immune challenge. Why were the time period change between the two experiments?

Line 325-327: I am wondering about the power of this analysis. P value are close to the significance level and might not completely capture the pattern. Using CHC distance using all CHC peaks might hide difference based on one or few peaks. I would advise the authors to confirm this absence of treatment effect using a random forest analysis (machine learning approach, see for instance Monnin et al. 2018 Journal of chemical ecology) to compare the three groups within each social environment and potentially identify some specific CHC driving the difference between the treatments.

Discussion

Line 333-335: I reckon that the only purpose of the first experiment was to provide evidence that the species studies display social immunity. However, the two main results are not discussed in the discussion. In eusocial insects, isolation by itself induces an important stress to the ant that could simply decrease their immune defence and hence their survival without necessary implying the existence of social immunity in the group. That allogrooming increases following infection is a more convincing evidence of social immunity. To my opinion, these two results might be not sufficient to claim without restriction that the authors demonstrated that the ant display a "classic" social immunity as stated by the authors in the discussion.

Decision letter (RSPB-2020-2570.R0)

29-Dec-2020

Dear Mr Alciatore:

I am writing to inform you that your manuscript RSPB-2020-2570 entitled "Individual immune status affects network position in a queenless ant" has, in its current form, been rejected for publication in Proceedings B.

This action has been taken on the advice of referees, who have recommended that substantial revisions are necessary. With this in mind we would be happy to consider a resubmission, provided the comments of the referees are fully addressed. However please note that this is not a provisional acceptance.

Sincerely,
 Professor Hans Heesterbeek
 mailto: proceedingsb@royalsociety.org

Associate Editor
 Board Member: 1

Comments to Author:

This manuscript has now been evaluated by three expert reviewers. All three reviewers agree that topically this manuscript is interesting. However, at least two of the three reviewers had questions about the novelty of the work. The reviewers point out, and I agree, that the more novel aspects of the paper are not well-developed. For example, first, the observation that individual animals (including many social insects) can detect infected conspecifics is widely reported in the literature. More interesting is how this variation in individual responses affects group-level behavior. Reviewer 1 suggests that greater emphasis on group-level hypotheses and results would enhance novelty. Second, the presence of social immunity in ants is clearly not unique. This paper focuses specifically on a clonal system, but as Reviewer 2 points out, the underlying rationale for why such systems should differ from classical systems is not provided, neither is there a robust discussion of what we can learn about social immunity from a clonal system that we couldn't from a non-clonal one. More generally, the context of this study needs to be better placed within the existing literature on both social immunity in social insects and behavioral defenses in vertebrates. This exercise would enhance novelty and reduce several instances of overstatement in the manuscript. In addition to the major novelty question, the reviewers also identified several other key issues associated with data interpretation (e.g. interpretation of tracking vs. behavioral data) and methods (e.g. the use of different ant genotypes for different experiments; treatment of non-independent network data; distinction between immune-challenged vs. exposed) that require careful consideration.

Reviewer(s)' Comments to Author:

Referee: 1

Comments to the Author(s)

Comments:

Overall, this is an interesting and timely study combining observations of behaviors and automated tracking to explore how queenless ants respond to fungus exposed/immune-challenged conspecifics. The main strength of the manuscript lies in (I) the multiple methods to expose/immune-challenge ants, (II) the comparative nature to previous studies in ants that focus on species with division of labor, and (III) the fact that group-wide observations were collected through automated movement tracking after the challenge experiments. I do, however, have concerns about certain interpretations of the manuscript. First, detection of infection risk is well

described in other study systems, and I think the main emphasis should be more on the fact that the authors collect group-level data. Second, the authors repeatedly discuss the results of tracking and behavioral observations of allogrooming interchangeably even though spatial associations and physical interactions should not be considered the same. Third, the differentiation between ants changing their behavior towards “exposed” and “immune-challenged” conspecifics needs to be clearer. Finally, I think the cost-benefit structure of caregiving could be explored in a bit more detail for both, fungal exposure and immune-challenges. Given these concerns, I would recommend major revisions before publishing the study. See below my specific comments.

Title:

I realize this is a personal preference, but I prefer descriptive titles that highlight the main result in more detail (e.g. in what way is network position affected?). “Immune status” also suggests to me that the immune status was measured instead of experimentally changed. This could be clarified by using “immune-challenged”.

Introduction:

General comments:

The introduction is well written, but I disagree slightly with the main points of the manuscript. First, detection of infection risk is well described in several species including humans (most prominently in the expression of avoidance responses). Experiments often focus on focal individuals or pairs of animals (I suggest adding some citations, see below). I think what makes this study somewhat novel is the fact that it integrates responses of multiple individuals simultaneously and focuses on their entire network. The authors mention this in lines 76-78 but I suggest emphasizing this more.

Second, the authors mainly emphasize the division of labor as a prominent driver of social immunity in their introduction, but also discussion. While I agree with the authors that one of the drivers of collective defenses in social insects seems to be division of labor and asymmetry in “importance” of individuals, high relatedness between individuals seems at least equally important. The authors state that pathogen exposure is potentially more costly for a clonal raider ant than for a sterile worker but if that clonal raider ant assists a cohort member that is genetically nearly identical, wouldn't these costs become negligible? To me, one novelty of the study lays in the fact that it is able to differentiate the two “conditions” of social immunity in social insects, (i) protecting more valuable individuals (such as the queen) and (ii) high relatedness, and that these behaviors are observed despite one is missing (division of labor). This could be clarified a bit more in the introduction (and discussion).

Minor comments:

Lines 39-40: I have not seen the term “social immunity” used broadly outside social insects where it describes a collective group defense (and includes seemingly altruistic behaviors) against pathogens that benefits the whole colony. I suggest either changing “many social species” to “many social insect species” or citing examples where social immunity and collective group defense is used in systems outside social insects.

Lines 46-47: Isn't there also the possibility of the spread of sublethal doses and hence, immunization within the colonies? This could be mentioned here, or somewhere else in the manuscript specifically in terms of the alleviated costs of continued allogrooming of exposed conspecifics.

Lines 63-64: Detection of infection risk has been studied in, for instance, mice, guppies, and mandrills, amongst others. In humans, this seems to be driven by a generalized “disgust-eliciting” response and a recent review outlines some of the key research nicely (Townsend et al, 2020, see below). The authors also mention studies in honeybees and ant pupae suggesting that detection of infection risk has been studied in social insects as well. I, therefore, do not agree with the statement that detection of infection risk is poorly understood. To clarify this, I would suggest

adding more citations to this section and emphasize the fact that the presented study focuses on whole-network responses rather than just detection itself.

Suggested citations:

Boillat et al 2008; The vomeronasal system mediates sick conspecific avoidance, *Current Biology*.
Stephenson et al. 2018; Transmission risk predicts avoidance of infected conspecifics in Trindiadian guppies. *Journal of Animal Ecology*.

Poirotte et al 2017; Mandrills use olfaction to socially avoid parasitized conspecifics. *Science Advances*.

Townsend et al 2020; Emerging infectious disease and the challenges of social distancing in human and non-human animals. *Proceedings B*.

Townsend et al 2020; Emerging infectious disease and the challenges of social distancing in human and non-human animals. *Proceedings B*.

Lines 67-70: As I understand this refers to a generalized disgust response. This has been discussed in a recent review by Townsend et. al. (see previous comment) which could be included as a citation. I would also suggest defining "slower" in this context. To me, if individuals can recognize general cues of infection (instead of relying on pathogen-specific cues), this could lead to a more rapid response.

Lines 82-85: For the broad readership of the target journal I suggest describing in a bit more detail how reproduction in this ant species works. How are these cohorts produced? Do individual ants clone themselves?

Lines 88-89: Please clarify minimal reproductive division of labor. Is there reproductive division of labor in this species to some extent?

Material and Methods:

Lines 113-120: I would suggest describing the infection-transmission dynamics of *M. robertsii*. It would be important to explain the difference between exposure and infection in this system (when does the fungus infect the ant? Is there a lag time between exposure and infection/physiological response? When do conspecifics respond to exposure versus infection on this timeline?).

Line 115: I suggest clarifying the term "germination check" for a broad readership.

Lines 124-125: Is there a reason why ants of different ages were used? Please explain.

Lines 136-138: How was determined whether age had a significant effect or not? This should be mentioned briefly.

Lines 139-140: Minor comment. At this point, I had to go back to check for the fixed factor descriptions above. I wonder if it would be helpful to include the actual factors in brackets such as treatment (infected/uninfected), social environment (alone/grouped)? The authors use a similar way of describing factors in their model in lines 146-148.

Line 144-145: Why is grooming received in minutes treated as a binomial response variable as opposed to a continuous response in a linear mixed effect model? Please clarify.

Line 146: I am confused about grooming received and the rounding of numbers. For instance, if an ant received 1.5 minutes in grooming, was this rounded up to 2 minutes? Why was the response variable rounded up to the next integer?

Lines 155-156: Was the volume of 0.1 ul used based on other studies? I also assume that, because these ants seem to be similar in morphology (and probably weight), there were no differences in the actual dosage each ant received.

Lines 156-159: Was survival of immune-challenged ants determined similarly to the fungal exposures above?

Lines 161-162: I noticed that ants for this assay were from genotype B and ants in the infection assay above were from genotype A. Are there significant differences between those genotypes or are they similar, please clarify this briefly at some point in the methods?

Lines 168-170: Defining social network edges requires deciding what minimum value of proximity or duration is sufficient for an association event. The authors define an isolation distance of >1mm (half a body length) and interaction distance of (< 1mm) and I appreciate the validation of the approach by observing the ants. A recent article observed effect sizes across a wide range of how an association is defined by re-sampling the collected data (Ripperger et al 2020, Behavioral Ecology). It would be fascinating to look at similar effects here, i.e. at which definition of an association does the effect disappear and what types of spatial associations are more or less important (with an emphasis on how pathogens spread, from close contact to further distance). I realize that this is beyond the scope of this study but might represent an interesting avenue for future research.

See for details:

Ripperger et al., 2020, Tracking sickness effects on social encounters via continuous proximity sensing in wild vampire bats. Behavioral Ecology.

Lines 174: I urge the authors to differentiate physical interactions (such as grooming) from spatial associations (distance to each other) more carefully. Being near to one another does not necessarily show that these ants groom each other.

Lines 208-213: As far as I understand here, the network characteristics (eigenvector centrality, strength, and skewness) are based on non-independent data. Please clarify if, and how you are dealing with non-independence in your analysis? Alternative approaches include node-label network permutations (see Ripperger et al, 2020, Behavioral Ecology).

Results:

Lines 253- 257: Instead of the mean difference and SEs, the authors could mention the means and bootstrapped confidence intervals for both treatment groups 0, 12, 18, and 48 hours post-exposure (and for other results throughout). I also recommend summarizing the means and bootstrapped confidence intervals shown in each plot in supplementary tables. I would suggest outlining the method of bootstrapping CIs at some point either in the results or the material and methods section (e.g. how often was the data resampled?).

Figure 2 (Lines 267-269): Do the authors have an observation for time = 0 (or shortly after the challenge) that could be added? If possible, I think this would be interesting to show because it would suggest more clearly that the physiological response of immune-challenged ants (delayed for about 9-15 hours, see Figure 3) increases the grooming response of conspecifics.

Discussion:

General comments:

>One of the interesting findings of the paper is that ants increase grooming of fungus-exposed conspecifics which could increase the exposed individual's survival. In this case, the cost-benefit structures seem to be less ambiguous. The exposed individual has a higher chance of survival (because of the removal of spores) and the groomer, even though at greater risk of infection, might even gain some personal and colony-level benefits through exposure-immunization as shown in other ant species. I would like to see more discussion about the finding that allogrooming towards immune-challenged ants also seems to increase (even though it might not increase their survival). How can this potentially benefit the immune-challenged ant and the ant that performs the grooming? Does this suggest that this care-taking behavior is a generalized

response and less dependent on the identity of the pathogen? What is the difference between exposure and “immune-challenged” in the cost-benefit balance for both the groomer and the receiver of grooming?

>The authors mainly highlight reproductive division of labor as an important difference between other social insect systems and their own. While I do believe this to be an important differentiation, I also think that the fact that all individuals are highly related to each other needs to be mentioned and discussed more explicitly. An individual might gain a lot from helping a nearly clonal conspecific in terms of its indirect fitness, especially if costs to itself are minor.

>While the authors mention their results of the automated tracking (Lines 359-363) I think it is important to differentiate here, and throughout the manuscript (see also methods part), that physical closeness (<1mm) does not equal physical interactions (i.e. grooming). In fact, networks constructed by using physical interactions such as grooming and spatial associations can differ quite a bit. For instance, an individual may be near a lot of others but does not receive grooming. Importantly, both can have profound effects on pathogen transmission which depends on the transmission pathway a pathogen takes. If a pathogen is transmitted through physical interactions (i.e. through grooming) a network constructed based on spatial associations might not predict its spread accurately. This could be discussed specifically.

Minor comments

Line 335-337: Do the authors have specific experiments in mind that could give us insights into the generality of collective defenses against diseases across group-living species? For instance, is it possible to vary kin relationship structure to some extent in these experimental colonies?

Lines 341-343: As I understand from the fungal pathogens used, an ant can be exposed (and infectious) but not infected yet (i.e. the fungus has not entered the body yet). It seems to me that ants could recognize exposure before the ant is infected (i.e. immune-challenged) and increase grooming. Could this be by detecting cues associated with the pathogen itself on the cuticle surface? In the immune-challenge experiment, however, recognition might be driven by behavioral/physiological changes related to the physiological response. This difference, if applicable, should be highlighted more explicitly.

Lines 348-350: The statement “the causal relationship between behavior and CHC profiles can go both ways” is confusing. Please revise this paragraph.

Lines 366-369: The authors mention that future work could determine whether this strategy plastically changes based on, for instance, risk adjustment of groomers. As I understand there are studies that show risk adjusted grooming in ants and I suggest citing them here.

Suggested citation:

Konrad et al 2018, Ants avoid superinfections by performing risk-adjusted sanitary care, PNAS.

Referee: 2

Comments to the Author(s)

In this study, the authors investigated the expression of social immunity in a queenless ant species. First, they show that workers survive better to infection with pathogens in presence of other workers compared to alone and that workers exposed to a pathogen received more allogrooming compared to non-infected workers. Then using both visual observations and an automatic tracking system, they show that immune-challenged workers are more often isolated in the nest, tend to be less active, as well as occupy a more central network position compared to control-challenged workers.

I love this topic and this system, and I was therefore very eager to read this manuscript.

Unfortunately, it mostly presents results that are already well known in other species, based on methods that have also been used earlier to test the same questions. So, I do not really understand the novelty of the presented work, and to what extent it should be of interest to a broad readership. In particular, it is unclear as to why clonal systems (which is the novel aspect of the work) should have completely different forms of social immunity (or no social immunity at all).

What are the predictions in terms of the proposed measurements (do less allogrooming? Be more active? Occupy different positions in the network?)? My take was that social immunity should be present there and in a not so much different form compared to the other ant systems... And this is exactly what you show. I do understand that it is generally worth having a look at many species to better understand complex processes such as social immunity, but I don't think that the associated results (if they do not emphasize major differences or induce major changes in the theory) can be of interest to the broad readership of Proc B. This is even clearer with the discussion, which is extremely short and does not put the results in a broad context. At the end of the reading, it is actually difficult to understand how the presented results provide novel and interesting insights into our general understanding of the evolution and expression of social immunity, or on how social species deal with infected group members. On other words, the discussion illustrates that this is a very solid but also very, very specific study that is mostly addressed to researchers looking for more taxonomically diverse examples of a system that is already well known. Overall, I strongly recommend the authors to rewrite their introduction to better emphasize the novelty of their work and provide clear rationales for the study, to (really) meat up their discussion to help readers that are not working on clonal ants to put the results into broad perspectives and then to submit their (solid!) work to a more specialised journal.

Below, you can find a few other comments:

Abstract: the abstract is oddly written, as it is difficult to understand what is the main question (and why it is important). For instance, it seems that the main question is (L25-26) "how this is achieved remains unclear", which is clearly not the main focus of the study.

L62: I think that this study (see below) could be cited here, as it provides support to this claim. Stroeymeyt, N., Grasse, A. V, Crespi, A., Mersch, D.P., Cremer, S., Keller, L., 2018. Social network plasticity decreases disease transmission in a eusocial insect. *Science* 362, 941–945. <https://doi.org/10.1126/science.aat4793>

L63-70: This paragraph comes a bit out of the blue and lacks references. However, paragraph L100-109 contains all the necessary references. I suggest combining these two parts into a single paragraph.

L78: I have the feeling that it is an overstatement, as at least one other study do this. (Stroeymeyt, N., Grasse, A. V, Crespi, A., Mersch, D.P., Cremer, S., Keller, L., 2018. Social network plasticity decreases disease transmission in a eusocial insect. *Science* 362, 941–945).

L111: The material and methods part is difficult to read, as it is particularly succinct and lacks any general description. For instance, what "pathogen exposure" stands for, as it contains survival assays (to test what?) and behavioural assays. This is the same for the "immune-challenges" part. Please, rewrite this entire part to facilitate its reading (first explain the general idea, and then provide the details).

L161: Is there any evidence that you can compare patterns obtained in genotype A (first part of the study) with patterns obtained in genotype B (this part) and with patterns obtained in genotype C (part 3 on CHCs)? Why did you use each genotype for each sperate experiment?

L243: I am afraid that the reported effect is not a clear demonstration of social immunity, as it may simply reflect that the stress of social isolation specifically hampers the immune response of isolated workers. This alternative interpretation receives support in gregarious earwigs: Kohlmeier, P., Holländer, K., Meunier, J., 2016. Survival after pathogen exposure in group-living insects: don't forget the stress of social isolation! *J. Evol. Biol.* 29, 1867–1872. <https://doi.org/10.1111/jeb.12916>

Referee: 3

Comments to the Author(s)

In this paper, the author nicely combine manual and automated behavioural analyses to test whether ants that are challenged by a non pathogenic elicitor change their behaviour in a way

that could affect social immunity such as allogrooming, activity, network position in the nest. They used a very interesting model system: the parthenogenetic and queenless clonal raider ant, *Ooceraea biroi*. This species offers the advantage to have workers genetically and morphologically homogeneous, that are totipotent but with no reproductive conflicts because of parthenogenetic reproduction. As a first step, the authors show that this peculiar species also displays social immunity as observed in other more classical ant species with division of labour. Ants exposed to pathogens (the classically used *Metarhizium* fungus) have a lower survival when they are kept alone than in small groups. They also show that pathogens exposed ants have higher rate of allogrooming. As a second step, they challenged a worker in a group of 9 ants and 5 larvae by a known elicitor of the insect immune system (*Saccharomyces cerevisiae* cell walls) that have no pathogenic effect, and compared its behaviours to a control worker of the same group (injected with just the PBS). They combine manual and automated behaviour tracking to compare allogrooming, activity, walking speed, isolation and network position. Immune challenged workers received more allogrooming, have a reduced activity level and have a more central network position and were less isolated than the control workers. The paper is nicely written, the analysis properly conducted.

Comments:

M&M

Line 129-130: what means old and young in terms of day old? How were the age of the ants known?

Genotypes: the genotypes used were different for each experiment (A for pathogene exposure, B for immune challenge and D for the analysis of CHC profil). Are there some specific reasons for using different genotypes? Do their lineages differ by some life history traits or behaviours that could affect the results?

Line 136-138: how was the effect of age tested? This should be said and the statistical results given to confirm this assertion (in a supplementary material if needed).

Line 145-146: not clear to me why the response variable "received grooming" should follow a binomial distribution. For a binomial model, the response variable should be either 0/1 or a number of responses over a number of trials. I guess that received grooming belong to the second type of response variable. In that case, the risk of over dispersion is important and is known to potentially affect the statistical significance of the factors if not taken into account. But may be I did not understand properly how was built the binomial model.

Line 156-158: in the immune challenge experiment, workers in the control-injected treatment also have their immune defence challenged as injection with PBS, by piercing the membrane can induce an immune response. This does not affect the conclusion of the paper, but just if the author would have also considered in their comparison a focal naïve worker, the effect of immune challenge could even have been more pronounced. Given that all workers were video tracked, I am wondering why the authors did not choose to also include a focal naïve worker in each group. This is particularly surprising given that for the study of CHC, the authors considered the three groups (immune challenged, control injected and naïve workers).

Line 160-162: did the authors have some experimental evidence that the behaviours of the ants in such small experimental groups of 9 ants and 5 larvae are representative of the behaviours display in full colonies? What is the mean size of colonies?

Results:

Fig1 and Fig2: I realised reading these two figures that the time post-treatment was not similar in the pathogen exposure experiment and the immune challenge. Why were the time period change between the two experiments?

Line 325-327: I am wondering about the power of this analysis. P value are close to the significance level and might not completely capture the pattern. Using CHC distance using all CHC peaks might hide difference based on one or few peaks. I would advise the authors to confirm this absence of treatment effect using a random forest analysis (machine learning approach, see for instance Monnin et al. 2018 Journal of chemical ecology) to compare the three groups within each social environment and potentially identify some specific CHC driving the difference between the treatments.

Discussion

Line 333-335: I reckon that the only purpose of the first experiment was to provide evidence that the species studies display social immunity. However, the two main results are not discussed in the discussion. In eusocial insects, isolation by itself induces an important stress to the ant that could simply decrease their immune defence and hence their survival without necessary implying the existence of social immunity in the group. That allogrooming increases following infection is a more convincing evidence of social immunity. To my opinion, these two results might be not sufficient to claim without restriction that the authors demonstrated that the ant display a "classic" social immunity as stated by the authors in the discussion.

Author's Response to Decision Letter for (RSPB-2020-2570.R0)

See Appendix A.

RSPB-2021-1456.R0

Review form: Reviewer 1

Recommendation

Accept as is

Scientific importance: Is the manuscript an original and important contribution to its field?

Excellent

General interest: Is the paper of sufficient general interest?

Excellent

Quality of the paper: Is the overall quality of the paper suitable?

Excellent

Is the length of the paper justified?

Yes

Should the paper be seen by a specialist statistical reviewer?

No

Do you have any concerns about statistical analyses in this paper? If so, please specify them explicitly in your report.

No

It is a condition of publication that authors make their supporting data, code and materials available - either as supplementary material or hosted in an external repository. Please rate, if applicable, the supporting data on the following criteria.

Is it accessible?

Yes

Is it clear?

Yes

Is it adequate?

Yes

Do you have any ethical concerns with this paper?

No

Comments to the Author

Dear Authors,

First, I would like to congratulate you to this interesting and timely study combining observations of behaviors and automated tracking to explore how queenless ants and their groups respond to exposed/infected conspecifics. I would also like to thank you for your careful and thorough revisions. I had initial concerns about the extent of the discussed concepts, statistical analysis, and the main emphasis of the manuscript, but you have addressed these thoroughly in your revisions. Since the initial review of this manuscript, several new reviews have discussed the cost-benefit balance and infection-induced behaviors across different taxa more broadly. While the article could potentially benefit from adding them if there is space for additional citations (especially in the first part of the introduction and in the discussion), it is not a must, and I am very happy with the revisions as is.

Potential citations:

- Hawley DM, Gibson AK, Townsend AK, Craft ME, Stephenson JF: Bidirectional interactions between host social behavior and parasites arise through ecological and evolutionary processes. *Parasitology* (2021)
- Stockmaier S, Stroeymeyt N, Shattuck EC, Hawley DM, Meyers LA, Bolnick DI: Infectious diseases and social distancing in nature. *Science* (2021)

Decision letter (RSPB-2021-1456.R0)

11-Aug-2021

Dear Ms Ulrich

I am pleased to inform you that your manuscript RSPB-2021-1456 entitled "Immune challenges increase network centrality in a queenless ant" has been accepted for publication in *Proceedings B*.

The referee has recommended publication, but also suggests some minor revisions to your manuscript. Therefore, I invite you to respond to the referee's comments and revise your manuscript. Because the schedule for publication is very tight, it is a condition of publication that you submit the revised version of your manuscript within 7 days. If you do not think you will be able to meet this date please let us know.

NB. From April 1 2013, peer reviewed articles based on research funded wholly or partly by RCUK must include, if applicable, a statement on how the underlying research materials – such

as data, samples or models – can be accessed. This statement should be included in the data accessibility section.

[http://datadryad.org/submit?journalID=RSPB&manu=\(Document not available\)](http://datadryad.org/submit?journalID=RSPB&manu=(Document%20not%20available)) which will take you to your unique entry in the Dryad repository. If you have already submitted your data to dryad you can make any necessary revisions to your dataset by following the above link. Please see <https://royalsociety.org/journals/ethics-policies/data-sharing-mining/> for more details.

Sincerely,
Professor Hans Heesterbeek
mailto:proceedingsb@royalsociety.org

Reviewer(s)' Comments to Author:

Referee: 1

Comments to the Author(s).

Dear Authors,

First, I would like to congratulate you to this interesting and timely study combining observations of behaviors and automated tracking to explore how queenless ants and their groups respond to exposed/infected conspecifics. I would also like to thank you for your careful and thorough revisions. I had initial concerns about the extent of the discussed concepts, statistical analysis, and the main emphasis of the manuscript, but you have addressed these thoroughly in your revisions. Since the initial review of this manuscript, several new reviews have discussed the cost-benefit balance and infection-induced behaviors across different taxa more broadly. While the article could potentially benefit from adding them if there is space for additional citations (especially in the first part of the introduction and in the discussion), it is not a must, and I am very happy with the revisions as is.

Potential citations:

- Hawley DM, Gibson AK, Townsend AK, Craft ME, Stephenson JF: Bidirectional interactions between host social behavior and parasites arise through ecological and evolutionary processes. *Parasitology* (2021)
- Stockmaier S, Stroeymeyt N, Shattuck EC, Hawley DM, Meyers LA, Bolnick DI: Infectious diseases and social distancing in nature. *Science* (2021)

Author's Response to Decision Letter for (RSPB-2021-1456.R0)

See Appendix B.

Decision letter (RSPB-2021-1456.R1)

13-Aug-2021

Dear Ms Ulrich

I am pleased to inform you that your manuscript entitled "Immune challenges increase network centrality in a queenless ant" has been accepted for publication in Proceedings B.

Data Accessibility section

Open Access

You are invited to opt for Open Access, making your freely available to all as soon as it is ready for publication under a CCBY licence. Our article processing charge for Open Access is £1700. Corresponding authors from member institutions (<http://royalsocietypublishing.org/site/librarians/allmembers.xhtml>) receive a 25% discount to these charges. For more information please visit <http://royalsocietypublishing.org/open-access>.

Paper charges

Sincerely,

Appendix A

Dear Prof. Heesterbeek,

Thank you for the opportunity to revise and resubmit our manuscript RSPB-2020-2570 to *Proceedings of the Royal Society B*. We thank the Reviewers and the Associate Editor for their constructive feedback and have carefully considered each of their comments. We believe that, as a result, our manuscript is greatly improved and considerably clearer. Below, we provide a brief summary of how we have addressed each of the primary concerns of the Reviewers as summarized by the Associate Editor, followed by our detailed responses to each of the Reviewers' comments.

Comments from all Reviewers and from the Academic Editor are reprinted below in plain text, followed by our responses in blue. Numbered references refer to the reference list at the end of this document. In addition to thoroughly responding to each comment, we list all associated changes to the manuscript along with the corresponding line numbers (line numbers refer to the clean copy of the revised manuscript unless otherwise specified). We also use tracked changes in the revised manuscript to indicate changes made in response to Reviewer and Academic Editor comments, for ease of identification.

We hope that, based on these revisions, you will find our manuscript suitable for publication in *Proceedings of the Royal Society B*.

Sincerely,

Yuko Ulrich (on behalf of all authors)

Associate Editor
Board Member: 1
Comments to Author:

This manuscript has now been evaluated by three expert reviewers. All three reviewers agree that topically this manuscript is interesting. However, at least two of the three reviewers had questions about the novelty of the work. The reviewers point out, and I agree, that the more novel aspects of the paper are not well-developed. For example, first, the observation that individual animals (including many social insects) can detect infected conspecifics is widely reported in the literature. More interesting is how this variation in individual responses affects group-level behavior. Reviewer 1 suggests that greater emphasis on group-level hypotheses and results would enhance novelty. Second, the presence of social immunity in ants is clearly not unique. This paper focuses specifically on a clonal system, but as Reviewer 2 points out, the underlying rationale for why such systems should differ from classical systems is not provided, neither is there a robust discussion of what we can learn about social immunity from a clonal system that we couldn't from a non-clonal one. More generally, the context of this study needs to be better placed within the existing literature on both social immunity in social insects and behavioral defenses in vertebrates. This exercise would enhance novelty and reduce several instances of overstatement in the manuscript. In addition to the major novelty question, the reviewers also identified several other key issues associated with data interpretation (e.g. interpretation of tracking vs. behavioral data) and methods (e.g. the use of different ant genotypes for different experiments; treatment of non-independent network data; distinction between immune-challenged vs. exposed) that require careful consideration.

We thank the Associate Editor for this clear summary. We here provide a summary of how we have addressed each of the major concerns.

1. **Novelty:** We now make it clearer in the manuscript that the main focus and novelty of our study lies in colony-wide network analyses following *immune-challenges* (as opposed to pathogen exposure). Few studies have quantified behavioral responses of social insects to immune-challenges and, to our knowledge, no study has so far done so in entire colonies. We believe that through changes in the Abstract, a better contextualisation of our study in the Introduction and a considerable expansion of the Discussion, the revised version of our manuscript now makes the focus, main contributions, and novelty of our study much clearer. See also our detailed replies to comments 1.2 and 1.6 below.

2. Social immunity: We do not claim that the existence of social immunity in the clonal raider ant is particularly surprising. Rather, and as noted by Reviewer 3 (comment 3.9), our experiments using actual pathogens were an important first step in our investigation, but in no way the core of our findings, which are presented in the rest of the study. Accordingly, we have refocused the Abstract, Introduction, and Discussion so that they put less emphasis on social immunity, and more emphasis on group-level behavioral responses to immune-challenges and their potential cost and benefits. We have also moved the figure on pathogen exposure experiments (formerly Fig. 1) to the Supplementary Material (Figs. S1-S2). See also our detailed reply to the main assessment of Reviewer 2 (pp. 10-11).
3. Methodology and data interpretation: we provide detailed replies to Reviewers' questions below and corresponding clarifications in the text on ant genotypes (comment 1.19; L. 139-142 in the Methods), the non-independence of network nodes (comment 1.22; L. 239 in the Methods), the distinction between social interactions and spatial associations (comments 1.21, 1.27; L. 399-405 in the Discussion), and the distinction between immune-challenged and pathogen-exposed ants (comment 1.25; L. 381-391 in the Discussion).

Addressing these main comments without exceeding article size limits required implementing other changes: including moving some figures to the Supplementary Material, merging two figures, reducing the number of references, and making the text more concise in parts.

Reviewer(s)' Comments to Author:

Referee: 1

Comments to the Author(s)

Comments:

Overall, this is an interesting and timely study combining observations of behaviors and automated tracking to explore how queenless ants respond to fungus exposed/immune-challenged conspecifics. The main strength of the manuscript lays (I) in the multiple methods to expose/immune-challenge ants, (II) the comparative nature to previous studies in ants that focus on species with division of labor, and (III) the fact that group-wide observations were collected through automated movement tracking after the challenge experiments. I do, however, have concerns about certain interpretations of the manuscript. First, detection of infection risk is well described in other study systems, and I think the main emphasis should be more on the fact that the authors collect group-level data. Second, the authors repeatedly discuss the results of tracking and behavioral observations of allogrooming interchangeably even though spatial associations and physical interactions should not be considered the same. Third, the differentiation between ants changing their behavior towards "exposed" and "immune-challenged" conspecifics needs to be clearer. Finally, I think the cost-benefit structure of caregiving could be explored in a bit more detail for both, fungal exposure and immune-challenges. Given these concerns, I would recommend major revisions before publishing the study. See below my specific comments.

We thank the Reviewer for the helpful and constructive comments.

Title:

1.1) I realize this is a personal preference, but I prefer descriptive titles that highlight the main result in more detail (e.g. in what way is network position affected?). "Immune status" also suggests to me that the immune status was measured instead of experimentally changed. This could be clarified by using "immune-challenged".

In response to this comment, we have changed the title to "Immune challenges increase network centrality in a queenless ant".

Introduction:

General comments:

1.2) The introduction is well written, but I disagree slightly with the main points of the manuscript. First,

detection of infection risk is well described in several species including humans (most prominently in the expression of avoidance responses). Experiments often focus on focal individuals or pairs of animals (I suggest adding some citations, see below). I think what makes this study somewhat novel is the fact that it integrates responses of multiple individuals simultaneously and focuses on their entire network. The authors mention this in lines 76-78 but I suggest emphasizing this more.

We have striven to more clearly outline the focus and novel aspects of the study in the revised version of the manuscript. We have modified the Introduction to: 1) refer to past work on the detection of infection risk (differentiating pathogen infections and immune-challenges) earlier in the text (2nd paragraph of the Introduction, L. 69), thereby making it clearer that we do not claim to be the first to investigate this topic, 2) emphasize that a contribution of our study is to investigate the effects of (specifically) immune challenges on the behavior of entire social groups, not only of focal individuals or pairs of animals (L. 84-91). See also our responses to comments 1.6 and 1.7 below for details on how we have improved the contextualization of our study.

1.3) Second, the authors mainly emphasize the division of labor as a prominent driver of social immunity in their introduction, but also discussion. While I agree with the authors that one of the drivers of collective defenses in social insects seems to be division of labor and asymmetry in “importance” of individuals, high relatedness between individuals seems at least equally important. The authors state that pathogen exposure is potentially more costly for a clonal raider ant than for a sterile worker but if that clonal raider ant assists a cohort member that is genetically nearly identical, wouldn't these costs become negligible? To me, one novelty of the study lays in the fact that it is able to differentiate the two “conditions” of social immunity in social insects, (i) protecting more valuable individuals (such as the queen) and (ii) high relatedness, and that these behaviors are observed despite one is missing (division of labor). This could be clarified a bit more in the introduction (and discussion).

We agree with the Reviewer. Both reproductive division of labor and high relatedness are thought to play an important role in promoting social immunity and were likely necessary for its initial evolution in social insects [1]. In the vast majority of social insects, both traits co-occur. The clonal raider ant is a rare social insect (to our knowledge, the only social insect) in which these two factors are fully unlinked (the former is absent, the latter almost maximal). It therefore allows to explore a new corner of the social “parameter space”, in a sense, and allows us to show that high relatedness (in the case of the clonal raider ant, extremely high relatedness) alone is sufficient for social immunity, even if reproductive division of labor is lost. In the original submission, we had focused our attention on reproductive division of labor because its absence is the most unusual trait (for a social insect) of the clonal raider ant (e.g., Cremer et al., 2018, argues that “social immunity, as originally defined for eusocial insects, is a derived trait that evolved when the unit of selection shifted from the individual to the colony, caused by the separation of germline and soma, which is an unconditional characteristic of superorganisms”, which could be interpreted as meaning that reproductive division of labor is necessary for social immunity). However, in response to this and other comments (1.26), we now better balance the discussion of reproductive division of labor and high relatedness both in the Introduction (L. 99-104) and Discussion (L. 414-419).

Minor comments:

1.4) Lines 39-40: I have not seen the term “social immunity” used broadly outside social insects where it describes a collective group defense (and includes seemingly altruistic behaviors) against pathogens that benefits the whole colony. I suggest either changing “many social species” to “many social insect species” or citing examples where social immunity and collective group defense is used in systems outside social insects.

We agree and rephrased this sentence so that “social immunity” is only used when referring to social insects (L. 41-43).

1.5) Lines 46-47: Isn't there also the possibility of the spread of sublethal doses and hence, immunization within the colonies? This could be mentioned here, or somewhere else in the manuscript specifically in terms of the alleviated costs of continued allogrooming of exposed conspecifics.

We now touch on the topic of immunization through allogrooming in the Discussion (L. 374-375), as part of an expanded discussion of the potential costs and benefits of allogrooming.

1.6) Lines 63-64: Detection of infection risk has been studied in, for instance, mice, guppies, and mandrills, amongst others. In humans, this seems to be driven by a generalized “disgust-eliciting” response and a recent review outlines some of the key research nicely (Townsend et al, 2020, see below). The authors also mention studies in honeybees and ant pupae suggesting that detection of infection risk has been studied in social insects as well. I, therefore, do not agree with the statement that detection of infection risk is poorly understood. To clarify this, I would suggest adding more citations to this section and emphasize the fact that the presented study focuses on whole-network responses rather than just detection itself.

Suggested citations:

Boillat et al 2008; The vomeronasal system mediates sick conspecific avoidance, *Current Biology*.

Stephenson et al. 2018; Transmission risk predicts avoidance of infected conspecifics in Trinidadian guppies. *Journal of Animal Ecology*.

Poirotte et al 2017; Mandrills use olfaction to socially avoid parasitized conspecifics. *Science Advances*.

Townsend et al 2020; Emerging infectious disease and the challenges of social distancing in human and non-human animals. *Proceedings B*.

We agree with the Reviewer that phenomenologically, the detection of infection risk has been demonstrated in a wide range of organisms. We thank the Reviewer for pointing us to relevant literature on this topic in vertebrates, which we have included as references in the Introduction (L. 43,75). We emphasize, however, that these studies focus on responses to the presence of *actually infected* animals, while the main focus of our study is on the ability to detect infection risk via immune activity (i.e., in the absence of actual pathogens), which is less well documented. This is an important distinction not only conceptually, but also practically, because any behavioral change (e.g., increased social interactions) in a host following actual infection could in principle and at least in part be a trait of the pathogen itself (e.g., virulence or parasite manipulation), not of the host. Only a handful of studies [2–7] have investigated behavioral responses to immune-challenges in social insects, and their results are equivocal (i.e. the results vary qualitatively across studies). Therefore, while the question has been studied before, it is far from being considered settled.

We now more clearly and specifically explain what we originally meant by “poorly understood” (and change the wording of that phrase at L. 27): while it is clear that the detection of infection risk is widespread across animal societies, both the mechanisms involved (visual vs. olfactory cues, pathogen-derived vs. immune-derived cues), the magnitude and direction of the behavioral responses to such detection (increase vs. decrease in social contacts), and the costs/benefits of such behavioral responses at the individual and group-levels vary considerably across systems. However, it can be difficult to compare results across studies, even for a given system, because studies use different approaches (e.g., live infections vs. immune-challenges, manual scoring of specific behaviors vs. automated tracking of all contacts, dyadic behavioral assays vs. group assays). Consequently, the results sometimes vary qualitatively, even within a system (e.g., effect of immune stimulation on social interactions in [2,3,7]). To our knowledge, we provide the first comprehensive study (combining live infections and immune-challenges, manual scoring of specific behaviors and automated tracking of all contacts, as well as chemical analyses) in a single, well-controlled biological system allowing comparisons across experiments within this study, but also across this and future studies.

1.7) Lines 67-70: As I understand this refers to a generalized disgust response. This has been discussed in a recent review by Townsend et. al. (see previous comment) which could be included as a citation. I would also suggest defining “slower” in this context. To me, if individuals can recognize general cues of infection (instead of relying on pathogen-specific cues), this could lead to a more rapid response.

While the detection of immune status relies on the ability to detect general cues of infection, we don’t think it can be characterized as a “generalized disgust response” in all animals, because it is unclear whether e.g. insects experience disgust, and because disgust would intuitively imply avoidance behavior, i.e. a decrease of social contacts with infectious individuals, while many studies (including ours) observe the opposite. However, we agree that the review by Townsend et al. is broadly relevant to the study, and we now cite it in the general part of the Introduction (L. 43). Following the Reviewer’s suggestion, we now also clarify our rationale for qualifying the detection of immune status as a slower mechanism than the direct detection of the pathogens at L. 76-79: because mounting an immune response takes time, detecting the immune status of social partners necessarily takes longer than detecting the pathogens directly (i.e., immediately after exposure). For example, allogrooming triggered by exposure to fungal spores is often immediate (as shown in our own study), but insects only mount an immune response after the spores have penetrated the cuticle, which typically takes 24-

48h, meaning that any behavioral response triggered by the detection of immune activation would be comparatively slow. Therefore, we do not mean “slower” in an evolutionary sense, but in a mechanistic sense.

1.8) Lines 82-85: For the broad readership of the target journal I suggest describing in a bit more detail how reproduction in this ant species works. How are these cohorts produced? Do individual ants clone themselves?

Individual workers indeed produce near clones of themselves by thelytokous parthenogenesis. We now include this information in the description of the clonal raider ant at L. 94-98.

1.9) Lines 88-89: Please clarify minimal reproductive division of labor. Is there reproductive division of labor in this species to some extent?

We use “minimal division of labor” to mean that in the clonal raider ant, unlike other ants, there is no subset of individuals (i.e., queens) that monopolize reproduction, but instead that all group members can and do reproduce (as in solitary animals, or even bacteria). Because small differences in reproductive output across individuals still likely exist (e.g., due to variation in age or small differences in morphology), this is not equivalent to “no reproductive division of labor”. In other words, reproductive skew is very low, but not necessarily zero in natural colonies of clonal raider ants. We have changed the wording of this sentence to “no strict division of labor” to clarify this point (L. 104).

Material and Methods:

1.10) Lines 113-120: I would suggest describing the infection-transmission dynamics of *M. robertsii*. It would be important to explain the difference between exposure and infection in this system (when does the fungus infect the ant? Is there a lag time between exposure and infection/physiological response? When do conspecifics respond to exposure versus infection on this timeline?).

We have included this information in the Material and Methods (L. 122-126): “Contact with the host cuticle induces *M. robertsii* conidiospores to germinate and penetrate the cuticle, typically within 24-48 hours. Thereafter, the fungus multiplies and spreads within the haemocoel, which at high conidiospore exposure doses results in host death within 3-7 days. The fungus then breaches the cuticle to form conidiospores on the cadaver’s surface, which can disperse to and infect new hosts.”

1.11) Line 115: I suggest clarifying the term “germination check” for a broad readership.

We agree and now clarify this term in the Material and Methods (L. 127-128), which now reads: “Prior to exposure, the conidiospore germination rate was assessed to be >95% by inoculating a standard Sabouraud dextrose agar plate and incubating at 23°C for 20 hrs.”

1.12) Lines 124-125: Is there a reason why ants of different ages were used? Please explain.

Originally, we had included age as a factor that might influence survival following pathogen exposure. However, as no statistical effect of age was found (see our response to the next comment), variation in age was not included in the following experiments.

1.13) Lines 136-138: How was determined whether age had a significant effect or not? This should be mentioned briefly.

Age was originally included in the survival statistical model as a fixed factor. Because it was not statistically significant, we deleted it from the final model and reported the reduced model in the original submission of this manuscript. Following the Reviewer suggestion, we now explain this rationale and provide the statistical results for age in the Material and Methods section (L. 157-159), which now reads “Age was originally included as a fixed factor in the model, but as it did not have a significant effect on survival ($\eta^2=0.77$, $Df=1$, $p=0.380$), it was removed from the model and variation in age was not included in subsequent experiments.”

1.14) Lines 139-140: Minor comment. At this point, I had to go back to check for the fixed factor descriptions above. I wonder if it would be helpful to include the actual factors in brackets such as treatment

(infected/uninfected), social environment (alone/grouped)? The authors use a similar way of describing factors in their model in lines 146-148.

We now include information about the factors included in all models in the Material and Methods (L. 160).

1.15) Line 144-145: Why is grooming received in minutes treated as a binomial response variable as opposed to a continuous response in a linear mixed effect model? Please clarify.

This was originally done for practical reasons, in this case to deal with challenges stemming from heavily zero-inflated data. However, prompted by this and other Reviewer comments (1.6, 3.4), we have revised our statistical analyses of grooming to use a continuous response variable in a generalized linear mixed effect model. Specifically, we now model our data (grooming received, in seconds) using a tweedie distribution, which allows us to both account for zero-inflation and treat our response variable as continuous, as described in the Material and Methods (L. 163-166). This change did not qualitatively affect our conclusions. Figs. 1 and S2, as well as the corresponding section of Results (L. 272-279, 281-289) have been updated accordingly.

1.16) Line 146: I am confused about grooming received and the rounding of numbers. For instance, if an ant received 1.5 minutes in grooming, was this rounded up to 2 minutes? Why was the response variable rounded up to the next integer?

We originally rounded the values because the binomial model used to analyze the data required integers. We decided to round all values up to avoid that grooming events lasting < 0.5 minute would be counted as no grooming (the resulting response variable could be described as the “fraction of observations (in minutes) in which at least some grooming was received by the focal ant”). However, as explained in response to the previous comment (1.15), we have revised our statistical analyses of grooming to use a continuous response variable, so that rounding of values is no longer required.

1.17) Lines 155-156: Was the volume of 0.1 ul used based on other studies? I also assume that, because these ants seem to be similar in morphology (and probably weight), there were no differences in the actual dosage each ant received.

The injection procedure was based on a previous study in the clonal raider ant [8]. The Reviewer is correct that clonal raider ants are very similar in size, and therefore the same injection volume was used for all individuals; however, some variation in injection volume unavoidably stemmed from the manual procedure. We now cite the above study in the Material and Methods section, provide more details regarding the injection procedure, and additionally make it clear that the injection volume was approximative (L.175-176).

1.18) Lines 156-159: Was survival of immune-challenged ants determined similarly to the fungal exposures above?

In both cases, survival was measured over the course of the experiment. For the immune-challenge experiment, this means survival was measured over 48h (vs. 13 days in the pathogen-exposure experiment). In the pathogen-exposure experiment, the goal was to test a specific hypothesis about survival, and we therefore analyzed the data with an explicit survival analysis. In the immune-challenge experiment, the goal was only to ensure that immune-challenges did not kill the animals, and the time-resolution of the survival data was low. We thus performed a simpler two-proportions z-test. We now provide information on the statistical approach at L. 177-180. Because of these differences in experimental design and purpose, we do not directly compare the survival results between the two experiments, but we can say with some confidence that over the timeframe of the experiments, immune-challenges did not increase ant mortality. Anecdotally, in separate projects, we have kept immune-challenged and control-injected ants in the lab for several weeks and both groups had similarly high long-term survival.

1.19) Lines 161-162: I noticed that ants for this assay were from genotype B and ants in the infection assay above were from genotype A. Are there significant differences between those genotypes or are they similar, please clarify this briefly at some point in the methods?

This question regarding ant genotypes was raised by all three Reviewers (comments 1.19, 2.6, and 3.2) highlighting an important point that requires clarification. We address all Reviewers' questions and comments regarding genotypes here.

Genotypes A, B, and D refer to genetic lineages with distinct mitochondrial haplotypes that were collected independently in the field [9]. All genotypes share the same basic life-history traits and characteristic features of this species (asexual and cyclic reproduction, phasic behavior). Both genetically and in terms of basic biology, genotypes of the clonal raider ant are not more different from one another than distinct colonies of sexually-reproducing social insects collected in the field (and which are customarily used in different experiments). While baseline differences between genotypes in e.g., reproductive rate [10] or CHC profiles [10] exist—as would be expected from different colonies in a sexually reproducing species—genotypes respond qualitatively similarly to the same experimental treatments (e.g., genotype A and B respond in the same way to changes in colony size, [11]). We can therefore assume that the conclusions drawn from experiments using different genotypes in this study are qualitatively generalizable across genotypes. The reason different genotypes were used in different experiments is practical. The different experiments presented in this study were performed at different times. In each experiment, we controlled variation in genotype and age by using ants from a single (or a small number of) age cohort(s), but each clonal raider ant colony only produces a new cohort every ca. 5 weeks. Therefore, the choice of genotype was based on the availability of colonies that were producing new cohorts of sufficient size in the lab at the time the experiment was planned. We now clarify in the Material and Methods section (L. 139-142) what we refer to as genotypes, and how these genotypes were chosen.

1.20) Lines 168-170: Defining social network edges requires deciding what minimum value of proximity or duration is sufficient for an association event. The authors define an isolation distance of >1mm (half a body length) and interaction distance of (< 1mm) and I appreciate the validation of the approach by observing the ants. A recent article observed effect sizes across a wide range of how an association is defined by re-sampling the collected data (Ripperger et al 2020, Behavioral Ecology). It would be fascinating to look at similar effects here, i.e. at which definition of an association does the effect disappear and what types of spatial associations are more or less important (with an emphasis on how pathogens spread, from close contact to further distance). I realize that this is beyond the scope of this study but might represent an interesting avenue for future research.

See for details:

Ripperger et al., 2020, Tracking sickness effects on social encounters via continuous proximity sensing in wild vampire bats. Behavioral Ecology.

We thank the Reviewer for this suggestion. We have read the careful methodological exploration of Ripperger et al. [12] (cited at L. 363, 367, albeit in a different context) with attention and it serves as a source of inspiration for our current work on social networks. We fully agree that exploring how experimental findings on social interactions depend on how an interaction is defined is both interesting, and crucial to gain biological insight into the consequences (positive and negative) of social interactions. This is a topic we plan to explore in future studies, for example when identifying the type and duration of social interactions that lead to the transmission of different pathogens, and which we believe will be facilitated by the ongoing development of image-based behavioral analyses (i.e., machine learning techniques to automatically detect different types of interactions, like grooming, aggression, or antennation). We now include some considerations on this topic in the Discussion (L. 392-405).

1.21) Lines 174: I urge the authors to differentiate physical interactions (such as grooming) from spatial associations (distance to each other) more carefully. Being near to one another does not necessarily show that these ants groom each other.

We are not sure whether this is a methodological or semantic point, and consider both possibilities below:

- Methodology: We use the term “physical interaction” to mean that two ants are in physical contact (i.e., that they touch each other). The (x, y) coordinates obtained from the tracking system are centered on the centroid of segmented ant silhouettes. As each ant measures 2-3 mm, if two single-ant centroids are at a distance of less than 1 mm from each other, this normally ensures that the two ants are close enough to each other to be in physical contact. However, this is not always the case. We now clarify this at L. 196-200: “Two ants were considered to be in physical contact if the centroids of their segmented silhouettes were <1 mm away from

each other. As clonal raider ants are 2-3 mm long, this cutoff ensured that segmented silhouettes were touching. However, segmented silhouettes can also contain more than two ants, in which case the physical contact between pairs of ants is not necessarily guaranteed". Because the tracking method is not perfect, we validated our approach manually: there were high correlations (Pearson correlation: $r: 0.84-0.99$, all $p < 1.333 \times 10^{-10}$) between contact networks based on manually-scored physical interactions and contact networks based on automatically-scored physical interactions (using the 1 mm cutoff), indicating that automated contact networks are a good proxy for networks of physical interactions.

- We fully agree that physical interactions as measured through automated tracking do not represent grooming (or any other specific type of social interaction), but they also represent more than mere spatial proximity, as we show they are a good proxy for actual physical contact. We don't think the term "physical interaction" as used at L. 194 is ambiguous in that respect. However, we interpret the comment as possibly meaning the Reviewer objects to the term "interaction" specifically and have therefore changed all instances of "physical interaction" to "physical contact". For consistency, we have replaced all instances of "interaction" (e.g., interaction network) with "contact" (e.g., contact network) in the revised version of the manuscript.

1.22) Lines 208-213: As far as I understand here, the network characteristics (eigenvector centrality, strength, and skewness) are based on non-independent data. Please clarify if, and how you are dealing with non-independence in your analysis? Alternative approaches include node-label network permutations (see Ripperger et al, 2020, Behavioral Ecology).

We understand the Reviewer's comment to refer to the fact that we compare individuals that belong to the same colony (e.g., immune-challenged and control-injected ants). We agree that these data points are non-independent, not only in network analyses, but in all analyses. In all analyses, we account for the non-independence of individual data statistically, by including colony as a random factor in statistical models (a common approach in social insect research, where individuals are rarely independent). We appreciate the careful methodological approach of Ripperger et al., which rigorously assesses how likely it is for a given pattern of social interactions to emerge due to chance in one bat social network. We would have used the node permutation approach used in that study if we had been studying a single ant social network. But we feel that the fact that we used 24 replicate networks alleviates the need for the node permutation approach. In other words, we test the hypothesis that the observed difference between immune-challenged ants and control-injected ants would have been observed by chance across 24 replicates. However, we recognize that how to best take into account the non-independence of network data is an actively debated topic [13], and if we have not addressed the comment with this reply, would be happy to further discuss this point with the Reviewer.

Results:

1.23) Lines 253- 257: Instead of the mean difference and SEs, the authors could mention the means and bootstrapped confidence intervals for both treatment groups 0, 12, 18, and 48 hours post-exposure (and for other results throughout). I also recommend summarizing the means and bootstrapped confidence intervals shown in each plot in supplementary tables. I would suggest outlining the method of bootstrapping CIs at some point either in the results or the material and methods section (e.g. how often was the data resampled?).

Following the Reviewer's advice, we substituted mean difference and SEs with means and bootstrapped confidence intervals throughout the text. We now provide more information about the bootstrapping procedure in the Material and Methods section (L. 153-154), as well as all means and bootstrapped confidence intervals in new supplementary tables (Tables S1-S3).

1.24) Figure 2 (Lines 267-269): Do the authors have an observation for time = 0 (or shortly after the challenge) that could be added? If possible, I think this would be interesting to show because it would suggest more clearly that the physiological response of immune-challenged ants (delayed for about 9-15 hours, see Figure 3) increases the grooming response of conspecifics.

Because injections are time-consuming (more so than pathogen exposure), we could only start recording videos ca. 6 hours post-injection. This means we do not have data from 0 hours post injection. However, while

we originally only included the manually tracked data starting at 12 hours in Fig 1, we have now manually scored behavior 6 hours post-injection (i.e., immediately after the start of recording), have added this new data point to Fig. 1, and have included it in the analysis. This narrows the time gap between pathogen-exposure and immune-challenge experiments and makes it clearer—like the Reviewer suggests—that the effect of immune-challenges indeed appears to be delayed compared to the effect of pathogen exposure.

Discussion:

General comments:

1.25) One of the interesting findings of the paper is that ants increase grooming of fungus-exposed conspecifics which could increase the exposed individual's survival. In this case, the cost-benefit structures seem to be less ambiguous. The exposed individual has a higher chance of survival (because of the removal of spores) and the groomer, even though at greater risk of infection, might even gain some personal and colony-level benefits through exposure-immunization as shown in other ant species. I would like to see more discussion about the finding that allogrooming towards immune-challenged ants also seems to increase (even though it might not increase their survival). How can this potentially benefit the immune-challenged ant and the ant that performs the grooming? Does this suggest that this care-taking behavior is a generalized response and less dependent on the identity of the pathogen? What is the difference between exposure and "immune-challenged" in the cost-benefit balance for both the groomer and the receiver of grooming?

We thank the Reviewer for raising these questions which we agree provide worthwhile avenues along which to expand our Discussion. We have now included a discussion of the costs and benefits of allogrooming under different scenarios in the Discussion (L. 362-380).

1.26) The authors mainly highlight reproductive division of labor as an important difference between other social insect systems and their own. While I do believe this to be an important differentiation, I also think that the fact that all individuals are highly related to each other needs to be mentioned and discussed more explicitly. An individual might gain a lot from helping a nearly clonal conspecific in terms of its indirect fitness, especially if costs to itself are minor.

This comment is similar to comment 1.3 and we refer the Reviewer to our detailed reply above. In short, we agree with the Reviewer that the high relatedness between nestmates probably largely counterbalances the (small) risk tied to performing altruistic grooming in the clonal raider ant. We now mention high relatedness more explicitly both in the Introduction (L. 102-103) and in the Discussion (L. 414-422).

1.27) While the authors mention their results of the automated tracking (Lines 359-363) I think it is important to differentiate here, and throughout the manuscript (see also methods part), that physical closeness (<1mm) does not equal physical interactions (i.e. grooming). In fact, networks constructed by using physical interactions such as grooming and spatial associations can differ quite a bit. For instance, an individual may be near a lot of others but does not receive grooming. Importantly, both can have profound effects on pathogen transmission which depends on the transmission pathway a pathogen takes. If a pathogen is transmitted through physical interactions (i.e. through grooming) a network constructed based on spatial associations might not predict its spread accurately. This could be discussed specifically.

This comment is similar to comment 1.21 and we refer the Reviewer to our detailed reply above. Briefly, we now use language that more clearly differentiates between social interactions (manual tracking) and physical contacts (automated tracking) throughout the text, and explicitly mention the distinction between these methods and corresponding data in the Discussion (L. 395-402).

Minor comments

1.28) Line 335-337: Do the authors have specific experiments in mind that could give us insights into the generality of collective defenses against diseases across group-living species? For instance, is it possible to vary kin relationship structure to some extent in these experimental colonies?

Varying kin structure in the clonal raider ant is possible and is done by creating (fully functional) mixed-genotype colonies in experiments [10,14]. However, mixed genotype colonies are not known to occur in nature, and it is therefore unclear whether clonal raider ants can be expected to have evolved plastic

adjustment of disease-relevant social behavior in response to variation in colony genetic structure. Rather than specific experiments in the clonal raider ant, we here mean that encompassing less conventional social insects (e.g., social insects without strict reproductive division of labor) and group-living species more generally, into a broader comparative view of social immunity might provide more robust insight into the conditions (e.g. relatedness, reproductive skew) that promote, or on the contrary hinder disease-relevant social behaviors, and could help bridge the gap between taxonomically-restricted subfields. We have slightly rephrased the sentence to make this clearer (L. 419-422).

1.29) Lines 341-343: As I understand from the fungal pathogens used, an ant can be exposed (and infectious) but not infected yet (i.e. the fungus has not entered the body yet). It seems to me that ants could recognize exposure before the ant is infected (i.e. immune-challenged) and increase grooming. Could this be by detecting cues associated with the pathogen itself on the cuticle surface? In the immune-challenge experiment, however, recognition might be driven by behavioral/physiological changes related to the physiological response. This difference, if applicable, should be highlighted more explicitly.

We thank the Reviewer for this comment. This interpretation captures exactly what we think is happening. Fungal spores are most likely detected directly (i.e., not via immune activation), as they can be detected (likely by chemical cues emanating from the fungus itself) on the ant's cuticle. Changes in the Introduction (L. 69-79), Methods (L. 122-126) and Discussion (L. 406-413) should make this clearer to the reader now.

1.30) Lines 348-350: The statement "the causal relationship between behavior and CHC profiles can go both ways" is confusing. Please revise this paragraph.

By that phrase, we meant that not only can changes in CHC profiles trigger behavioral changes in other individuals, but social interactions with other individuals (e.g., grooming) can also affect CHC profiles (e.g., if social partners deposit CHC onto the cuticle by grooming). This is now stated more explicitly at L. 409-411.

1.31) Lines 366-369: The authors mention that future work could determine whether this strategy plastically changes based on, for instance, risk adjustment of groomers. As I understand there are studies that show risk adjusted grooming in ants and I suggest citing them here.

Suggested citation:

Konrad et al 2018, Ants avoid superinfections by performing risk-adjusted sanitary care, PNAS.

We agree with the Reviewer. We now mention this type of risk adjustment as part of our expanded discussion of the cost and benefits of grooming (L. 367-375) and we discuss and cite the suggested study at L. 372.

Referee: 2

Comments to the Author(s)

In this study, the authors investigated the expression of social immunity in a queenless ant species. First, they show that workers survive better to infection with pathogens in presence of other workers compared to alone and that workers exposed to a pathogen received more allogrooming compared to non-infected workers. Then using both visual observations and an automatic tracking system, they show that immune-challenged workers are more often isolated in the nest, tend to be less active, as well as occupy a more central network position compared to control-challenged workers.

I love this topic and this system, and I was therefore very eager to read this manuscript. Unfortunately, it mostly presents results that are already well known in other species, based on methods that have also been used earlier to test the same questions. So, I do not really understand the novelty of the presented work, and to what extent it should be of interest to a broad readership. In particular, it is unclear as to why clonal systems (which is the novel aspect of the work) should have completely different forms of social immunity (or no social immunity at all). What are the predictions in terms of the proposed measurements (do less allogrooming? Be more active? Occupy different positions in the network?)? My take was that social immunity should be present there and in a not so much different form compared to the other ant systems... And this is exactly what you show. I do understand that it is generally worth having a look at many species to better understand complex processes such as social immunity, but I don't think that the associated results (if they do not emphasize major differences or induce major changes in the theory) can be of interest to the broad readership of Proc B. This is even clearer with the discussion, which is extremely short and does not put the

results in a broad context. At the end of the reading, it is actually difficult to understand how the presented results provide novel and interesting insights into our general understanding of the evolution and expression of social immunity, or on how social species deal with infected group members. On other words, the discussion illustrates that this is a very solid but also very, very specific study that is mostly addressed to researchers looking for more taxonomically diverse examples of a system that is already well known. Overall, I strongly recommend the authors to rewrite their introduction to better emphasize the novelty of their work and provide clear rationales for the study, to (really) meet up their discussion to help readers that are not working on clonal ants to put the results into broad perspectives and then to submit their (solid!) work to a more specialised journal.

This general assessment, along with some comments from other Reviewers, helped us realise that the focus, main contributions, and novelty of our study did not come across sufficiently clearly in the original version of the manuscript. Reviewer 2 here focuses on our finding that clonal raider ants display social immunity, but this is in no way the primary contribution of our work. We were not surprised to find that the clonal raider ant shows social immunity but showing that it does was in our opinion—as Reviewer 3 points out in comment 3.9—an important first step to ground the rest of the experiments on a solid basis, and something we felt was necessary to address given the unusual biology of our study system (compared to “standard” ants—see also our reply to comment 1.3 above). This first result is therefore not the core of our manuscript, but more of a “sanity check” that allows us to progress to our main question, which has to do with the ability of social insects to detect the *immune status* of their nestmates, and the resulting individual and group-level behavioral responses. Accordingly, the results on social immunity are now discussed only briefly in the Introduction and Discussion, and the figure showing these results has been moved to the Supplementary Material. Conversely, we put more emphasis on the focus of our study, and its novelty. Few studies have studied the ability of social insects to detect the immune status of their nestmates [2–7], and to our knowledge, no study has so far done so at the colony level (e.g. by measuring interactions between all colony members). We also provide the first comprehensive study (combining live infections and immune-challenges, manual scoring of specific behaviors and automated tracking of all contacts, as well as chemical analyses) of responses to infection risk in a single, well-controlled biological system with high replicability. We believe that through the structural changes in the Abstract and Introduction, as well as a considerable expansion of the Discussion, as recommended by the Reviewer, the revised version of our manuscript now makes the focus, contributions, and novelty of our study much clearer.

Below, you can find a few other comments:

2.1) Abstract: the abstract is oddly written, as it is difficult to understand what is the main question (and why it is important). For instance, it seems that the main question is (L25-26) “how this is achieved remains unclear”, which is clearly not the main focus of the study.

In light of this and other comments, we have considerably modified the Abstract (and removed the quoted phrase) to clarify the main focus and contribution of our study.

2.2) L62: I think that this study (see below) could be cited here, as it provides support to this claim. Stroeymeyt, N., Grasse, A. V., Crespi, A., Mersch, D.P., Cremer, S., Keller, L., 2018. Social network plasticity decreases disease transmission in a eusocial insect. *Science* 362, 941–945. <https://doi.org/10.1126/science.aat4793>

The study suggested by the Reviewer is relevant to our study, and it is therefore cited multiple times, but we do not believe it provides empirical evidence for this particular statement. Specifically, our sentence at L.64 reads: “Alternatively, a colony could in principle gain the benefits of allogrooming while limiting the colony-wide transmission risk associated with increased social contacts by skewing the social interactions of infectious colony members towards a reduced set of social partners (e.g., specialized caregivers).” In our understanding, the study by Stroeymeyt et al. does not show increases in allogrooming, or a reduction in the number of social partners.

2.3) L63-70: This paragraph comes a bit out of the blue and lacks references. However, paragraph L100-109 contains all the necessary references. I suggest combining these two parts into a single paragraph.

We thank the Reviewer for that suggestion. We have included the relevant references from the second paragraph in the first paragraph (now starting at L. 69). We keep the information about CHCs in a separate paragraph, as we delve into social insect-specific details there that we feel are less appropriate for a general paragraph on infection detection mechanisms across systems.

2.4) L78: I have the feeling that it is an overstatement, as at least one other study do this. (Stroeymeyt, N., Grasse, A. V, Crespi, A., Mersch, D.P., Cremer, S., Keller, L., 2018. Social network plasticity decreases disease transmission in a eusocial insect. *Science* 362, 941–945).

To be clear, we do not claim to be the first to use automated tracking on entire colonies of social insects; while relatively new, automated tracking has by now been used in dozens of studies on social insects (including the study pointed out by the Reviewer). As is clear from its context, the sentence now starting at L. 84 was specifically about using automated tracking to study the effects of immune-challenges (“While the effects of immune-challenges with various immune elicitors on the insect immune system are well-characterized, few studies have quantified their effects on social behaviour in social insects, and the reported behavioural effects are based on focal-individual approaches that do not identify the social partners and thus provide limited information on colony-level responses. To alleviate this problem, we use automated tracking to monitor individual behaviour and social interactions of all colony members”, where “this problem” referred to the previous sentence, which is explicitly about immune challenges). The study the Reviewer refers to deals with actual pathogen exposure (and importantly, pathogen exposure that has not yet induced an immune response), which most likely occurs through a different mechanism (direct pathogen detection). We do not question that the proposed citation is highly relevant to our study, and we accordingly cite it in several instances in our study, but we do not feel it is appropriate in this particular context. To avoid any confusion, we rephrased the sentence, which now reads: “Here, we use automated tracking to measure both individual behaviour and patterns of interactions between all colony members. This allows us to evaluate how immune-challenges affect the network position of challenged individuals and the colony’s network structure over time.”

2.5) L111: The material and methods part is difficult to read, as it is particularly succinct and lacks any general description. For instance, what "pathogen exposure" stands for, as it contains survival assays (to test what?) and behavioural assays. This is the same for the "immune-challenges" part. Please, rewrite this entire part to facilitate its reading (first explain the general idea, and then provide the details).

More methodological details on fungal spore preparation, spore germination, and ant genotype choice were added at L. 126-127, L. 127-128, and L.140-142 respectively. We now also explicitly state the general goal of each subsection at its beginning (e.g., “To assess the effects of immune-challenges on individual-level and colony-level behavior, [...]”, “To study behavioural responses to pathogen-exposed nestmates, [...]”) before describing the practical details.

2.6) L161: Is there any evidence that you can compare patterns obtained in genotype A (first part of the study) with patterns obtained in genotype B (this part) and with patterns obtained in genotype C (part 3 on CHCs)? Why did you use each genotype for each separate experiment?

We refer the Reviewer to our detailed reply to the same question raised by Reviewer 1 in comment 1.19.

2.7) L243: I am afraid that the reported effect is not a clear demonstration of social immunity, as it may simply reflect that the stress of social isolation specifically hampers the immune response of isolated workers. This alternative interpretation receives support in gregarious earwigs:

Kohlmeier, P., Holländer, K., Meunier, J., 2016. Survival after pathogen exposure in group-living insects: don't forget the stress of social isolation! *J. Evol. Biol.* 29, 1867–1872. <https://doi.org/10.1111/jeb.12916>

The study referred to by the Reviewer shows that earwigs reared in isolation both before and after exposure to fungal spores have similarly low mortality than individuals reared in groups both before and after exposure, but significantly higher mortality than individuals kept in groups *before* exposure and alone *after* exposure. In other words, that study shows that in earwigs, it is the change in social environment (sudden isolation), not the absence of social partners *per se* that induces mortality in pathogen-exposed individuals. While we agree that this rigorous experimental design shows the importance of factors that had previously been overlooked,

experiments involving long-term social isolation (here, the earwigs were kept alone for 5 weeks pre-treatment) cannot be performed in ants, which rapidly die if kept alone [15]. Therefore, if the experimental design of [16] is set as the standard to demonstrate social immunity, social immunity cannot be demonstrated in eusocial insects, the taxon in which it has originally been defined [17]. If the Reviewer has suggestions for alternative (i.e., practically possible) experiments that could be performed to convince them that social immunity indeed exists in the clonal raider ant, as in other ants, we would be happy to consider them.

More importantly, in our study (and in many other studies on eusocial insects), the support for social immunity stems not only from survival data (as in the earwig study), but from quantitative behavioral data. We show that workers that are exposed to otherwise lethal doses of fungal spores receive intense grooming from their nestmates (Fig. S2) and have dramatically improved survival. While we do not claim that grooming was the only factor affecting survival in our experiments, and while the social context might have affected immune function in our experiments as well, we believe that a reasonable case can be made that the dramatic increase in grooming observed had an impact on the dramatic decrease in mortality, and therefore, that the behavior of groomers helped their nestmates survive. Indeed, there is a strong mechanistic basis for this effect: allogrooming is known to physically remove infectious spores from the body surface of ants [18,19], and this is known to reduce mortality [20–22]. Thus, while we agree that survival data alone might not have been sufficient to demonstrate social immunity, the combined survival and behavioral data, taken in the context of the vast body of literature in social insects justifies talking about social immunity in the clonal raider ant.

Nonetheless, in response to this and another Reviewer's comment (3.9 below), we have toned down the corresponding sentence in the Discussion (L. 416), which now reads: "Here, we show that an ant with maximal relatedness but minimal reproductive division of labour and therefore homogeneous value across individuals displays disease-relevant social behaviour (survival-enhancing allogrooming) that is qualitatively similar to the forms of social immunity found in "standard" social insects ", so that the reader can decide for themselves whether or not this indeed constitutes social immunity. We also removed any mention of social immunity from the Abstract in the process of refocusing it on our main question.

Referee: 3

Comments to the Author(s)

In this paper, the author nicely combine manual and automated behavioural analyses to test whether ants that are challenged by a non pathogenic elicitor change their behaviour in a way that could affect social immunity such as allogrooming, activity, network position in the nest. They used a very interesting model system: the parthenogenetic and queenless clonal raider ant, *Ooceraea biroi*. This species offers the advantage to have workers genetically and morphologically homogeneous, that are totipotent but with no reproductive conflicts because of parthenogenetic reproduction. As a first step, the authors show that this peculiar species also displays social immunity as observed in other more classical ant species with division of labour. Ants exposed to pathogens (the classically used *Metarhizium* fungus) have a lower survival when they are kept alone than in small groups. They also show that pathogens exposed ants have higher rate of allogrooming. As a second step, they challenged a worker in a group of 9 ants and 5 larvae by a known elicitor of the insect immune system (*Saccharomyces cerevisiae* cell walls) that have no pathogenic effect, and compared its behaviours to a control worker of the same group (injected with just the PBS). They combine manual and automated behaviour tracking to compare allogrooming, activity, walking speed, isolation and network position. Immune challenged workers received more allogrooming, have a reduced activity level and have a more central network position and were less isolated than the control workers. The paper is nicely written, the analysis properly conducted.

Comments:

M&M

3.1) Line 129-130: what means old and young in terms of day old? How were the age of the ants known?

The ants were 210 and 30 days old, respectively. We now include this information more explicitly in the text (L. 137-138). Additionally, the description of the clonal raider ant biology has been expanded to better explain how discrete cohorts of individuals of known age are produced (L. 94-98).

3.2) Genotypes: the genotypes used were different for each experiment (A for pathogen exposure, B for immune challenge and D for the analysis of CHC profil). Are there some specific reasons for using different genotypes? Do their lineages differ by some life history traits or behaviours that could affect the results?

The same question was raised by Reviewer 1 in comment 1.19 and we refer the Reviewer to our detailed reply above.

3.3) Line 136-138: how was the effect of age tested? This should be said and the statistical results given to confirm this assertion (in a supplementary material if needed)

We refer the Reviewer to our detailed reply to the same question raised by Reviewer 1 in comment 1.13. The relevant information is now provided in the Methods (L. 157-159).

3.4) Line 145-146: not clear to me why the response variable “received grooming” should follow a binomial distribution. For a binomial model, the response variable should be either 0/1 or a number of responses over a number of trials. I guess that received grooming belong to the second type of response variable. In that case, the risk of over dispersion is important and is known to potentially affect the statistical significance of the factors if not taken into account. But may be I did not understand properly how was built the binomial model.

A similar comment was made by Reviewer 1 (comment 1.15), and we refer Reviewer 3 to our detailed reply above. Briefly, we have revised our statistical analyses of grooming to use a continuous response variable in a generalized linear mixed effect model. Specifically, we now model our data using a tweedie distribution, which allows us to both account for zero-inflation and treat our response variable as continuous, as described in the Methods (L. 163-166). This change does not affect our conclusions. Figures 1 and S2, as well as the corresponding section of the Results (L. 272-279, 281-289) have been updated accordingly.

3.5) Line 156-158: in the immune challenge experiment, workers in the control-injected treatment also have their immune defence challenged as injection with PBS, by piercing the membrane can induce an immune response. This does not affect the conclusion of the paper, but just if the author would have also considered in their comparison a focal naïve worker, the effect of immune challenge could even have been more pronounced. Given that all workers were video tracked, I am wondering why the authors did not choose to also include a focal naïve worker in each group. This is particularly surprising given that for the study of CHC, the authors considered the three groups (immune challenged, control injected and naïve workers).

We decided to focus on the comparison between immune-challenged and control-injected ants in our analyses because this is the most appropriate comparison in our opinion (as it fully controls for the experimental manipulation, including the associated stress). Accordingly, we only showed the corresponding data in the original submission. However, we agree that including the data from naive ants adds a layer of information that might be of interest to readers. We have updated Fig. 2 to include behavioral tracking data obtained from naive ants. As the Reviewer predicted, an effect of the injection itself is apparent, with the control-injected ants being intermediate between immune-challenged ants and naive ants in several cases (Figs. 2B,C,F). We now describe this pattern in the Results section (L. 334-337). However, we refrain from including the naive ants in the statistical analyses, because the potential number of pairwise comparisons (3 pairwise tests for each of 9 timepoints) becomes unreasonable.

In addition to the inclusion of naive ants, Fig. 2 was also modified to account for the fact that the first and last time windows were shorter than the others (3 hours instead of 6 hours). Where necessary (i.e., for isolation, activity, and strength, Fig. 2 A-B, G), we now correct for variation in time-window duration and express the behavioral metrics as a proportion of time, instead of absolute time.

3.6) Line 160-162: did the authors have some experimental evidence that the behaviours of the ants in such small experimental groups of 9 ants and 5 larvae are representative of the behaviours display in full colonies? What is the mean size of colonies?

Clonal raider ant colonies collected in the field range from a dozen to a few hundred workers [9,23,24], i.e., are typically somewhat larger than colonies used in our experiments. While the experimental colonies used here are at the lower size range or somewhat smaller than colonies in the field, previous experimental work has established that small laboratory colonies of ca. 10 workers have high fitness (indistinguishable from colonies

orders of magnitude larger) and display complex collective behavior, including stable division of labor [11] and stereotypical group raiding behavior [25]. We chose the group sizes used here based on this previous work. We have added a section in the Methods (L. 147-148) to clarify these points.

Results:

3.7) Fig1 and Fig2: I realised reading these two figures that the time post-treatment was not similar in the pathogen exposure experiment and the immune challenge. Why were the time period change between the two experiments?

This issue was also raised by Reviewer 1 in comment 1.24. Because injections are time-consuming (more so than pathogen exposure), we could only start recording videos 6 hours post-injection in the immune-challenge experiment. This means we do not have videos immediately after the treatment (0h post-injection). However, while we originally only analyzed data starting at 12h post-injection in Fig. 1, we have now additionally manually scored behavior at 6h post-injection (i.e., immediately after the start of recording), have added this new data point to Fig. 1, and have included it in the analysis.

3.8) Line 325-327: I am wondering about the power of this analysis. P value are close to the significance level and might not completely capture the pattern. Using CHC distance using all CHC peaks might hide difference based on one or few peaks. I would advise the authors to confirm this absence of treatment effect using a random forest analysis (machine learning approach, see for instance Monnin et al. 2018 Journal of chemical ecology) to compare the three groups within each social environment and potentially identify some specific CHC driving the difference between the treatments.

Thank you for this comment. We originally also performed a random forest analysis but decided not to include it as its results were qualitatively similar to the ones from the main analysis presented in the manuscript. For completeness, we now include the random forest analysis in the supplementary material (Fig. S6-S8) and refer to it in the Results section at L. 344-345.

Discussion

3.9) Line 333-335: I reckon that the only purpose of the first experiment was to provide evidence that the species studies display social immunity. However, the two main results are not discussed in the discussion. In eusocial insects, isolation by itself induces an important stress to the ant that could simply decrease their immune defence and hence their survival without necessarily implying the existence of social immunity in the group. That allogrooming increases following infection is a more convincing evidence of social immunity. To my opinion, these two results might be not sufficient to claim without restriction that the authors demonstrated that the ant display a "classic" social immunity as stated by the authors in the discussion.

The same comment was raised also by Reviewer 2 in comment 2.7 and we refer Reviewer 3 to our detailed answer above. In response to both comments, we toned down the corresponding sentence in the Discussion (L. 416: "Here, we show that an ant with maximal relatedness but minimal reproductive division of labour and therefore homogeneous value across individuals displays disease-relevant social behaviour (survival-enhancing allogrooming) that is qualitatively similar to the forms of social immunity found in "standard" social insects ") so that the reader can decide for themselves whether or not this indeed constitutes social immunity. We also took out any mention of social immunity from the Abstract in the process of refocusing it on our main question.

The wording of this comment, however, also suggests that there might have been a misunderstanding about our experimental design. We wish to clarify that the effect of social isolation was accounted for by analysing the survival of sham-exposed ants kept in isolation. If isolation *per se* (i.e., independently from infection status) reduced survival in our experiment, we would expect isolated sham-exposed ants to have lower survival than sham-exposed ants kept in groups. Instead, their survival was indistinguishable from that of ants kept in groups (solid orange line in Fig. S1; pairwise comparison: Sham alone vs Sham Grouped, $p=0.560$).

References:

1. Cremer S, Pull CD, Fürst MA. 2018 Social Immunity: Emergence and Evolution of Colony-Level Disease Protection. *Annu. Rev. Entomol.* **63**, 105-123. (doi:10.1146/annurev-ento-020117-043110)
2. Geffre AC *et al.* 2020 Honey bee virus causes context-dependent changes in host social behavior. *Proc. Natl. Acad. Sci. U.S.A.* **117**(19), 10406–13. (doi:10.1073/PNAS.2002268117)
3. Richard F-J, Aubert A, Grozinger CM. 2008 Modulation of social interactions by immune stimulation in honey bee, *Apis mellifera*, workers. *BMC Biol.* **6**, 50. (doi:10.1186/1741-7007-6-50)
4. Aubert A, Richard FJ. 2008 Social management of LPS-induced inflammation in *Formica polyctena* ants. *Brain Behav. Immun.* **22**, 833–837. (doi:10.1016/j.bbi.2008.01.010)
5. Richard FJ, Holt HL, Grozinger CM. 2012 Effects of immunostimulation on social behavior, chemical communication and genome-wide gene expression in honey bee workers (*Apis mellifera*). *BMC Genomics* **13**, 558. (doi:10.1186/1471-2164-13-558)
6. Conroy TE, Holman L. 2021 Social immunity and chemical communication in the honeybee: immune-challenged bees enter enforced or self-imposed exile. *bioRxiv.*, 2020.08.21.262113. (doi:10.1101/2020.08.21.262113)
7. Kazlauskas N, Klappenbach M, Depino AM, Locatelli FF. 2016 Sickness Behavior in Honey Bees. *Front. Physiol.* **7**, 1–10. (doi:10.3389/fphys.2016.00261)
8. Chandra V, Fetter-Pruneda I, Oxley PR, Ritger AL, McKenzie SK, Libbrecht R, Kronauer DJC. 2018 Social regulation of insulin signaling and the evolution of eusociality in ants. *Science* **361**, 398-402. (doi:10.1126/science.aar5723)
9. Tribble W, McKenzie SK, Kronauer DJC. 2020 Globally invasive populations of the clonal raider ant are derived from Bangladesh. *Biol. Lett.* **16**, 20200105. (doi:10.1098/rsbl.2020.0105)
10. Teseo S, Châline N, Jaisson P, Kronauer DJC. 2014 Epistasis between adults and larvae underlies caste fate and fitness in a clonal ant. *Nat. Commun.* **5**. (doi:10.1038/ncomms4363)
11. Ulrich Y, Saragosti J, Tokita CK, Tarnita CE, Kronauer DJC. 2018 Fitness benefits and emergent division of labour at the onset of group living. *Nature* **560**, 635-638. (doi:10.1038/s41586-018-0422-6)
12. Ripperger SP, Stockmaier S, Carter GG. 2020 Tracking sickness effects on social encounters via continuous proximity sensing in wild vampire bats. *Behav. Ecol.* **31**, 1296–1302. (doi:10.1093/beheco/araa111)
13. Hart JDA, Weiss MN, Brent LNJ, Franks DW. 2021 Common Permutation Methods in Animal Social Network Analysis Do Not Control for Non-independence. *bioRxiv.*, 2021.06.04.447124. (doi:10.1101/2021.06.04.447124)
14. Ulrich Y, Kawakatsu M, Tokita CK, Saragosti J, Chandra V, Tarnita CE, Kronauer DJC. 2020 Emergent behavioral organization in heterogeneous groups of a social insect. *bioRxiv.*, 2020.03.05.963207. (doi:10.1101/2020.03.05.963207)
15. Koto A, Mersch D, Hollis B, Keller L. 2015 Social isolation causes mortality by disrupting energy homeostasis in ants. *Behav. Ecol. Sociobiol.* **69**, 583–591. (doi:10.1007/s00265-014-1869-6)
16. Kohlmeier P, Holländer K, Meunier J. 2016 Survival after pathogen exposure in group-living insects: don't forget the stress of social isolation! *J. Evol. Biol.* **29**, 1867-72. (doi:10.1111/jeb.12916)
17. Cremer S, Armitage SAO, Schmid-Hempel P. 2007 Social Immunity. *Curr. Biol.* **17**, 693–702. (doi:10.1016/j.cub.2007.06.008)
18. Oi DH, Pereira RM. 1993 Ant Behavior and Microbial Pathogens (Hymenoptera: Formicidae). *Fla. Entomol.* **76**, 63. (doi:10.2307/3496014)

19. Reber A, Purcell J, Buechel SD, Buri P, Chapuisat M. 2011 The expression and impact of antifungal grooming in ants. *J. Evol. Biol.* **24**, 954–964. (doi:10.1111/j.1420-9101.2011.02230.x)
20. Graystock P, Hughes WOH. 2011 Disease resistance in a weaver ant, *Polyrhachis dives*, and the role of antibiotic-producing glands. *Behav. Ecol. Sociobiol.* **65**, 2319–2327. (doi:10.1007/s00265-011-1242-y)
21. Bos N, Kankaanpää-Kukkonen V, Freitak D, Stucki D, Sundström L. 2019 Comparison of Twelve Ant Species and Their Susceptibility to Fungal Infection. *Insects* **10**, 271. (doi:10.3390/insects10090271)
22. Theis FJ, Ugelvig LV, Marr C, Cremer S. 2015 Opposing effects of allogrooming on disease transmission in ant societies. *Philos. Trans. R. Soc. Lond. B Biol. Sci.* **370**. (doi:10.1098/rstb.2014.0108)
23. Tsuji K, Yamauchi K. 1995 Production of females by parthenogenesis in the ant, *Cerapachys biroi*. *Insectes Soc.* **42**, 333–336. (doi:10.1007/BF01240430)
24. Ravary F, Jaisson P. 2002 The reproductive cycle of thelytokous colonies of *Cerapachys biroi* Forel (Formicidae, Cerapachyinae). *Insectes Soc.* **49**, 114–119.
25. Chandra V, Gal A, Kronauer DJC. 2021 Colony expansions underlie the evolution of army ant mass raiding. *Proc. Natl. Acad. Sci. U.S.A.* **118**. (doi:10.1073/pnas.2026534118)

Appendix B

Dear Prof. Heesterbeek,

Thank you for your rapid handling of our manuscript RSPB-2021-1456 and your decision to publish it in *Proceedings of the Royal Society B: Biological Sciences*.

We also thank the Reviewer for their positive feedback and further suggestions for improvements. We now cite the two suggested references in the Introduction of our manuscript.

We are submitting a photograph of a colony of clonal raider ants, our study system, for consideration as a cover image. We confirm that we have full permission to reproduce the image (taken by one of the authors of the study) online and in print in perpetuity.

Sincerely,

Yuko Ulrich (on behalf of all authors)

Reviewer(s)' Comments to Author:

Referee: 1

Comments to the Author(s).

Dear Authors,

First, I would like to congratulate you to this interesting and timely study combining observations of behaviors and automated tracking to explore how queenless ants and their groups respond to exposed/infected conspecifics. I would also like to thank you for your careful and thorough revisions. I had initial concerns about the extent of the discussed concepts, statistical analysis, and the main emphasis of the manuscript, but you have addressed these thoroughly in your revisions. Since the initial review of this manuscript, several new reviews have discussed the cost-benefit balance and infection-induced behaviors across different taxa more broadly. While the article could potentially benefit from adding them if there is space for additional citations (especially in the first part of the introduction and in the discussion), it is not a must, and I am very happy with the revisions as is.

Potential citations:

- Hawley DM, Gibson AK, Townsend AK, Craft ME, Stephenson JF: Bidirectional interactions between host social behavior and parasites arise through ecological and evolutionary processes. Parasitology (2021)*
- Stockmaier S, Stroeymeyt N, Shattuck EC, Hawley DM, Meyers LA, Bolnick DI: Infectious diseases and social distancing in nature. Science (2021)*

We thank the Referee for this positive assessment and for the additional suggested references, which we have included in the Introduction at L. 42, 44, and 85. We sincerely appreciate the constructive feedback we have received throughout the review process, which we feel has greatly improved our manuscript.